# Viral rhodopsins 1 are an unique family of light-gated cation channels

Dmitrii Zabelskii [ID] et al.[#]

Phytoplankton is the base of the marine food chain as well as oxygen and carbon cycles and thus plays a global role in climate and ecology. Nucleocytoplasmic Large DNA Viruses that infect phytoplankton organisms and regulate the phytoplankton dynamics encompass genes of rhodopsins of two distinct families. Here, we present a functional and structural characterization of two proteins of viral rhodopsin group 1, OLPVR1 and VirChR1. Functional analysis of VirChR1 shows that it is a highly selective, $Na^+/K^+$-conducting channel and, in contrast to known cation channelrhodopsins, it is impermeable to $Ca^{2+}$ ions. We show that, upon illumination, VirChR1 is able to drive neural firing. The 1.4 Å resolution structure of OLPVR1 reveals remarkable differences from the known channelrhodopsins and a unique ion-conducting pathway. Thus, viral rhodopsins 1 represent a unique, large group of light-gated channels (viral channelrhodopsins, VirChR1s). In nature, VirChR1s likely mediate phototaxis of algae enhancing the host anabolic processes to support virus reproduction, and therefore, might play a major role in global phytoplankton dynamics. Moreover, VirChR1s have unique potential for optogenetics as they lack possibly noxious $Ca^{2+}$ permeability.

---

[#]A list of authors and their affiliations appears at the end of the paper.

Microbial and animal rhodopsins (type-1 and 2 rhodopsins, respectively) comprise a superfamily of heptahelical (7-TM) transmembrane proteins covalently linked to a retinal chromophore[1,2]. Type-1 rhodopsins are the most abundant light-harvesting proteins that have diverse functions, such as ion pumping, ion channeling, sensory, and enzymatic activities[3–11]. The discovery, in 2000, of the light-driven pump proteorhodopsin (PR) in marine microbes triggered an extensive search of metagenomes for light-activated proteins[12]. As a result, about 10,000 rhodopsin genes have been identified in archaea, bacteria, unicellular eukaryotes, and viruses, although the biological functions of most of these proteins remain elusive. The study of microbial rhodopsins culminated in the discovery of channelrhodopsins[13] yielding the development of optogenetics, the revolutionary method for controlling cell behavior in vivo using light-gated channels and light-driven pumps[13,14]. Currently, major efforts are being undertaken to identify rhodopsins with novel functions and properties that could be harnessed to enhance optogenetic applications[15–18].

In 2012, bioinformatic analysis of proteins encoded by nucleocytoplasmic large DNA viruses (NCLDV) resulted in the identification of highly-diverged PR homologs (hereafter, viral rhodopsins) in Organic Lake phycodnavirus and *Phaeocystis globosa* viruses, which belong to the extended Mimiviridae family[19]. Phylogenetic analysis shows that, within the rhodopsin superfamily, viral rhodopsins form a monophyletic family that consists of two distinct groups, VR1 and VR2[20]. Recently, a DTS-rhodopsin from the VR1 group (VirR$_{DTS}$) has been reported to pump protons when expressed in *E. coli* plasma membrane[21]. Almost at the same time, the OLPVRII protein from the VR2 group has also been shown to have a proton-pumping capacity, although forming unusual pentamers in lipid membrane[22]. The broad representation of a distinct family of rhodopsins in virus genomes implies an important light-dependent function in virus-host interactions, but the nature of this function remains uncertain. Given that NCLDV play a major role in marine algae population dynamics, elucidation of the virus-host interaction mechanisms would make an important contribution to a better understanding of the impact of viruses on global ecology and climate[23,24].

Here we present the results of a comprehensive structure-function study of two homologous proteins (61% sequence similarity) from the VR1 group, OLPVR1 (ADX06642.1), and VirChR1 (TARA-146-SRF-0.22-3-C376786_1). We show that unlike previously reported data[21], viral rhodopsins of the VR1 group demonstrate sodium/potassium-selective channelrhodopsin activity when expressed in human neuroblastoma cells. Upon light absorption, VirChR1 depolarizes cell membranes to a level sufficient to drive neural firing. This finding, together with the fact that, in contrast to the previously characterized channelrhodopsins, VirChR1 is not permeable for calcium ions, suggested that viral rhodopsins of the VR1 group (VirChR1s) could become invaluable optogenetic tools for the remote control of $Ca^{2+}$-dependent processes in the cell without $Ca^{2+}$-induced noxious side effects. To verify this, we expressed VirChR1 in neurons and showed directly that the channel is able to elicit firing. Following functional characterization, we crystallized and solved multiple structures of OLPVR1 that revealed unique structure-function features of VirChR1s.

The recent hypothesis of proton pumping by VirR$_{DTS}$ (from the same phylogenetic VR1 group) was based on pH measurements in suspensions of *E. coli* cells expressing VirR$_{DTS}$ in their plasma membrane[21]. Taking into account that such experiments cannot definitively prove the absence of ion channel properties of rhodopsin, here we directly addressed the hypothesis that VirR$_{DTS}$ is a light-gated channel. We expressed VirR$_{DTS}$ in

HEK293 cells, conducted voltage-clamp measurements, and demonstrated that VirR$_{DTS}$ is also a light-gated ion channel.

Thus, taking into account all the data, high homology of VirChR1s, high conservation of functionally key amino acids, we suggest that the VirChR1s (including VirR$_{DTS}$ from giant PgV virus infecting *Phaeocystic*, an important phytoplankton component) form a so far undescribed group of light-gated channels different from the known channelrhodopsins.

## Results

**Metagenomic search for viral rhodopsins genes and sequence analysis.** To obtain a comprehensive set of rhodopsins in the vast metagenomic sequence database produced by the *Tara Ocean* project, we compared 36 rhodopsin sequences representative of the previously identified major groups to the sequences of all open reading frames from *Tara Ocean* contigs. This search retrieved 2584 Type 1 rhodopsins, of which 385 belonged to VR1 and 172 belonged to VR2 as confirmed by phylogenetic analysis that also supported the monophyly of viral rhodopsins family (Fig. 1a and Supplementary Fig. 1). Consistent with the monophyly of viral rhodopsins and the separation of the VR1 and VR2 groups, the examination of sequence alignments detected several amino acid motifs that partially differed between the two groups. The amino acid triad implicated in proton exchange with the retinal Schiff base (residues 85, 89, and 96 in the reference bacteriorhodopsin[25,26]) had the form DTS/DTT in the VR1 group and DTT/DSV in the VR2 group. The members of the VR1 group are characterized by several fully conserved residues, such as S11, Q15, E51, K192, N193, N197, and N205 (annotated with OLPVR1 numbering, Supplementary Fig. 2) that are mainly located in proximity to the retinal Schiff base. Despite the overall low structural similarity with chlorophyte cation-conducting channelrhodopsins (Fig. 1d), VR1 rhodopsins retain the two highly conservative glutamates in TM2 (E44 and E51 in OLPVR1 corresponding to E83 and E90 in *Cr*ChR2) that have been shown to be critical for *Cr*ChR2 ion channelling[27,28]. In addition, as it will be shown later, the VR1 group has a signature topological feature, namely, an extended TM4 helix that consists of its transmembrane (TM4) and membrane-associated parts (ICL2) and has not been previously observed in characterized microbial rhodopsins (Fig. 1e and Supplementary Fig. 3).

**Spectroscopic characterization of VirChR1s.** To characterize photochemical characteristics of viral channelrhodopsins, we expressed C-terminally his-tagged OLPVR1 and VirChR1 proteins in *E. coli* and purified them using combination of Ni-NTA and size-exclusion chromatography methods (see "Methods" for details). VirChR1 protein was additionally supplemented with BRIL protein on the N-terminus of the protein, to improve expression level of the protein[29]. Both OLPVR1 and VirChR1 show absorption spectra sensitive to blue light with $\lambda_{max}$ of 500 nm and 507 nm, respectively at pH 7.5 (Fig. 2a), which is consistent with the fact that blue light penetrates deep throughout the seawater photic zone[30]. Similar to VirR$_{DTS}$ rhodopsin[21], *Hs*BR[31] and proteorhodopsins[32], OLPVR1 and VirChR1 undergo a ~30 nm spectral red-shift under acidic conditions, associated with protonation of retinal chromophore counterion (Fig. 2b, c). The Schiff base region of VirChR1s is reminiscent of those in light-driven proton pumps, such as *Hs*BR, suggesting that D76 in OLPVR1 (D80 in VirChR1) acts as counterion, as in *Hs*BR (Fig. 2d). In order to estimate the pKa values we plotted the absorption maximum values against buffer pH and fitted the data by the Henderson-Hasselbalch equation with a single pKa (Fig. 2e). The resulting pKa values for OLPVR1 (pKa = 4.8) and VirChR1 (pKa = 3.5) are in a good agreement with pKa = 3.6,

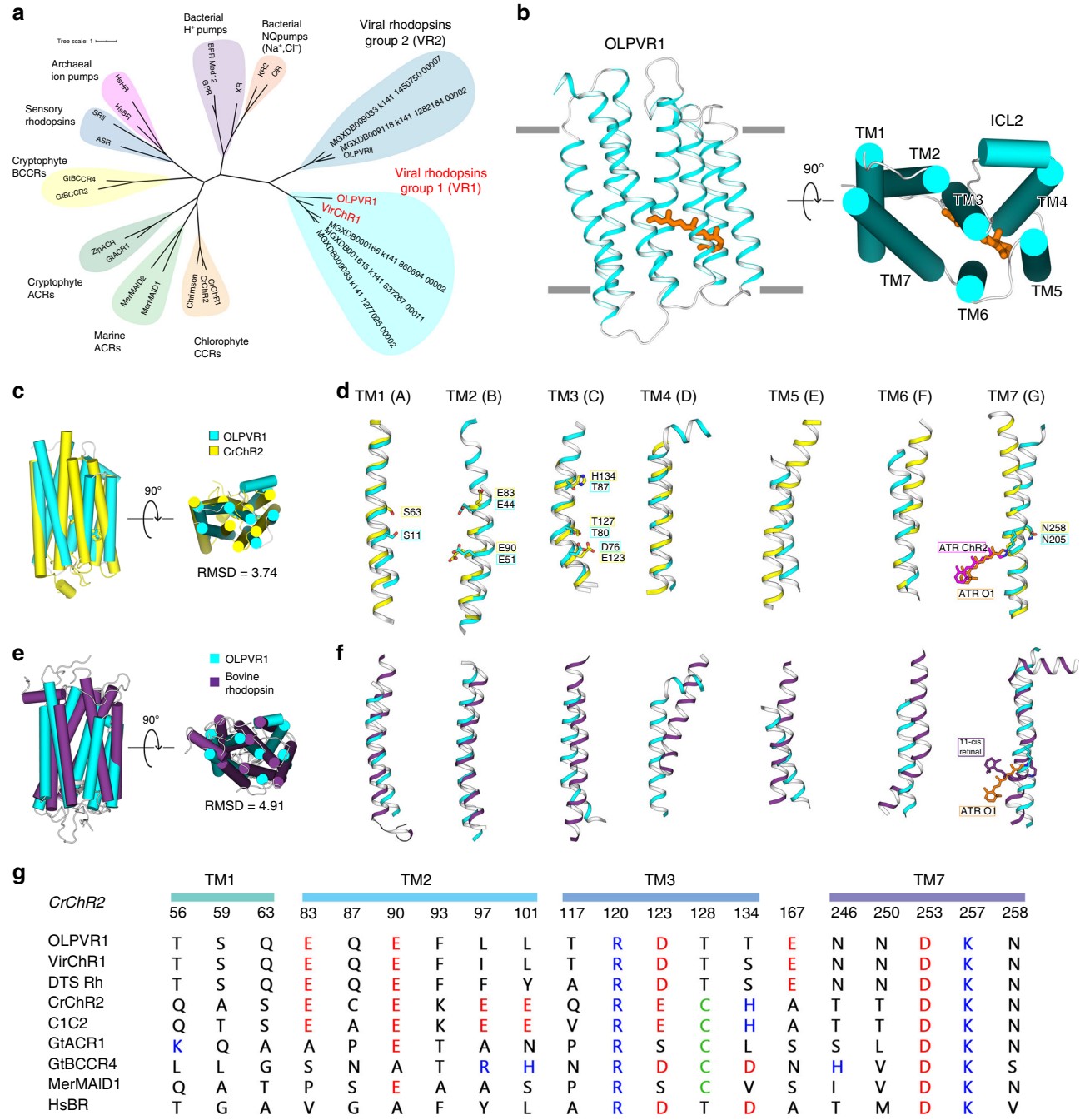

**Fig. 1 Phylogenetic and structural overview of the viral channelrhodopsins group. a** Unrooted phylogenetic tree of the channelrhodopsin superfamily, including viral channelrhodopsin representatives. Scale bar indicates the average number of amino acid substitutions per site. CCR, cation-conducting channelrhodopsin, ACR, anion-conducting channelrhodopsin. OLPVR1 and VirChR1 proteins are additionally indicated in red. Rhodopsins were named according to the previous works[18,101,102]. **b** Crystal structure of OLPVR1 protein, viewed parallel to membrane (left), and from the extracellular side (right). All-*trans* retinal (ATR) is depicted with orange sticks. The hydrophobic membrane boundaries were calculated with the PPM server and are shown by gray lines[103]. **c** Structure alignment of OLPVR1 and *Cr*ChR2 (PDB ID: 6EID[104]) structures colored cyan and yellow, respectively. RMSD, root mean square deviation of atomic positions. **d** Individual TM helices are shown after the superimposition of the OLPVR1 and *Cr*ChR2 rhodopsins. **e** Structure alignment of OLPVR1 and bovine rhodopsin (PDB ID: 1U19[43]) structures colored cyan and purple, respectively. **f** Individual TM helices are shown after the superimposition of the OLPVR1 and bovine rhodopsin. **g** Alignments of functionally important residues of transmembrane helices 1, 2, 3, and 7 of representative proteins from channelrhodopsin subfamilies. Positively and negatively charged residues are highlighted blue and red; cysteine residues are highlighted green.

previously reported for VirR$_{DTS}$ rhodopsin[21]. The one-unit difference between OLPVR1 and VirChR1 pKa values might be possibly explained by difference in relative positions of the TM1–3 and TM7 helices and difference in neighboring to

counterion residues, such as I50 and L79 in OLPVR1, which are replaced with V53 and I83 in VirChR1 (Supplementary Fig. 2).

To elucidate photocycle kinetics of viral channelrhodopsins, we performed transient absorption measurements with OLPVR1-

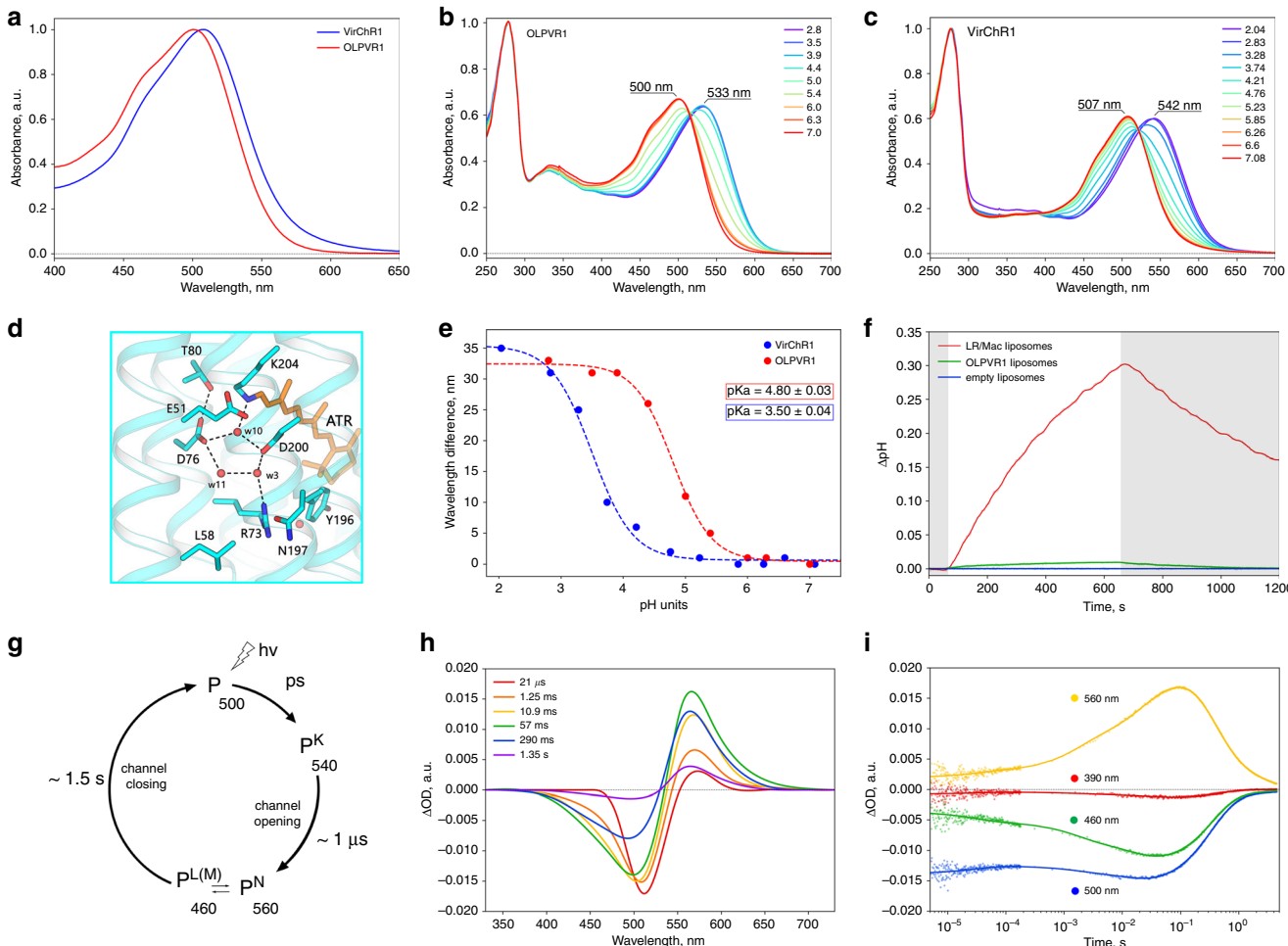

**Fig. 2 Spectroscopic characterization of viral channelrhodopsins. a** Normalized absorption spectra of OLPVR1 and VirChR1 proteins at neutral pH (pH 7.5). **b**, **c** Absorption spectra of OLPVR1 and VirChR1 at acidic (pH 2.1–7.1) pH range, normalized for absorption at 280 nm. **d** Schiff base region of OLPVR1 protein, key residues and water molecules are shown as sticks and spheres; hydrogen bonds are shown as dashed lines. **e** Absorption spectra of OLPVR1 at acidic (pH 2.8–7.0) pH range, normalized for absorption at 280 nm. **e** Red shift of UV-visible absorption spectrum and protonation of counterion of OLPVR1 and VirChR1. Wavelength maximum values are shown as circles. Sigmoidal curve fits are presented as dashed lines. The $pK_a$ values were calculated using a sigmoidal fit. **f** Ion-transport activity assay of OLPVR1-containing proteoliposomes in 100 mM NaCl salt. The onset of illumination is indicated with white (light on) and gray (light off) background color, pH was adjusted to pH 6.0 prior to measurements. LR/Mac-containing liposomes and empty liposomes were used as positive and negative controls, respectively. **g** Schematic model of viral rhodopsins photocycle. **h** Transient absorption spectra and **i** time traces at specific wavelengths of wild type OLPVR1 protein at pH 7.5.

containing nanodiscs that revealed three distinct intermediate states of OLPVR1 photocycle (Fig. 2g). An early decaying K-like state ($\lambda_{max} = 540$ nm), followed by major accumulation of L-like ($\lambda_{max} = 460$ nm) and N-like ($\lambda_{max} = 560$ nm) states that live for about 1.5 s (Fig. 2h, i). Unlike other channelrhodopsins, OLPVR1 lacks a detectable M-state that is generally associated with the ion-conducting state of the protein (Fig. 2c, red curve). At the same time, VirR$_{DTS}$ also forms similar intermediates and lacks the M-like state[21]. Although the equilibrium between the L-like and N-like states is the major candidate for the ion-conducting state in VirChR1s, further investigations are required for its identification.

**Functional analysis of OLPVR1**. To investigate the functional properties of the VR1 group, we performed the measurements of the pH changes in the suspension of proteoliposomes containing the viral rhodopsin, upon continuous light illumination. This method allows determination of the pumping activity and is often used for the characterization of microbial rhodopsins[7,33–35]. We did not observe any significant ion-translocation ability of the

viral rhodopsin OLPVR1 in pH change experiments with proteoliposomes (Fig. 2g). Under continuous bright light illumination, OLPVR1-containing liposomes did not show any substantial pH change of the external solvent (Fig. 2f). The maximum pH shift of the proteoliposomes suspension containing OLPVR1 (0.03 pH units), is about 10 times less than of that containing the eukaryotic light-driven proton pumping rhodopsin from *Leptosphaeria maculans* (LR/Mac[36]) (Fig. 2f). Taking into account the known fact that microbial rhodopsins in liposomes are usually oriented in the opposite direction to that in the cell membrane[7,33,34], we conclude that OLPVR1 possesses a weak outward proton pumping activity. The outward proton pumping has also been shown for another member of the VR1 group, VirR$_{DTS}$[21]. However, in the case of OLPVR1, the pumping activity was much lower than that of VirR$_{DTS}$. Importantly, although the pH recordings upon light illumination using a reduced system, such as proteoliposomes suspension, are suitable for the validation of the light-driven ion pumps (bacteriorhodopsin, proteorhodopsin, halorhodopsin and bacterial ion pumps), they cannot demonstrate the ion-channeling activity of

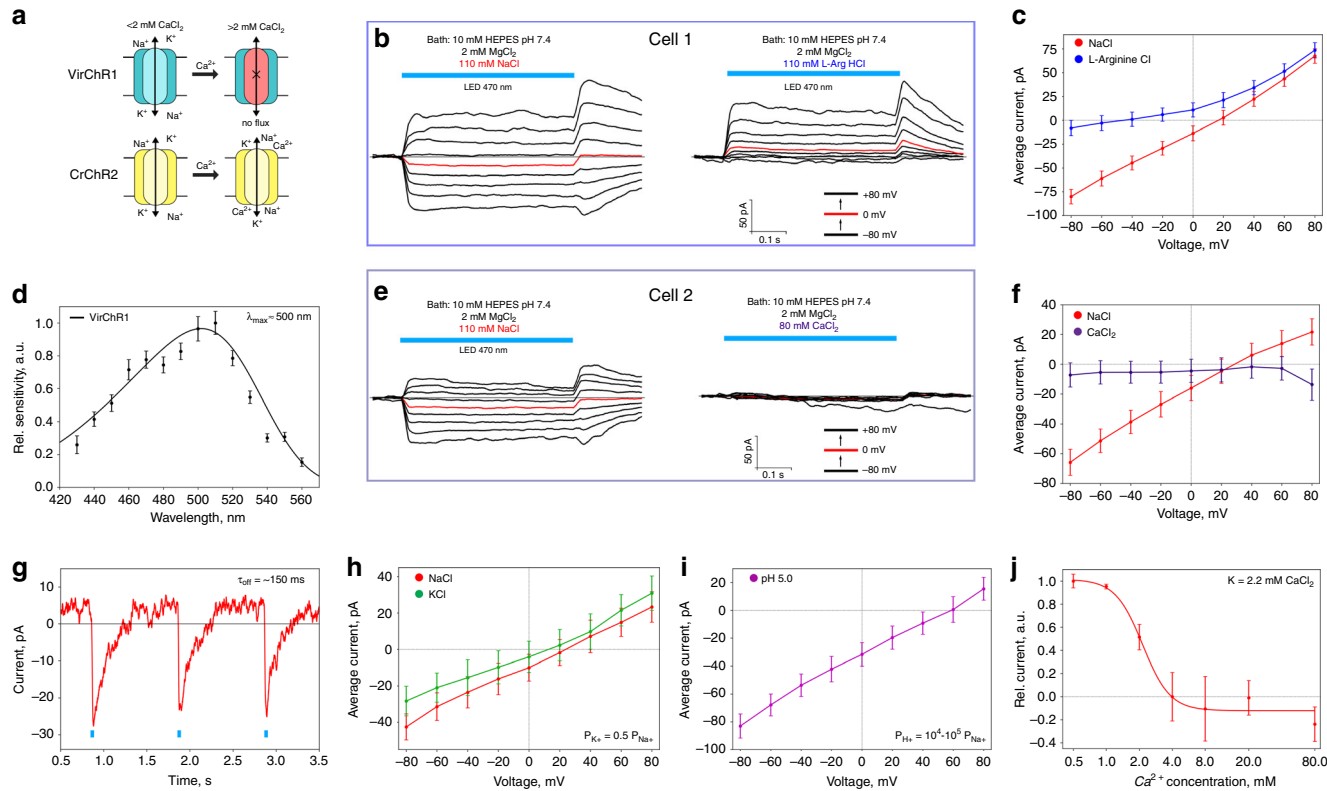

**Fig. 3 Ion selectivity and physiological features of VirChR1. a** Schematic comparison of VirChR1 and *Cr*ChR2 ion channeling activity under different calcium concentrations, membrane boundaries are shown schematically as black horizontal lines. **b** Voltage-clamp records from $n = 1$ representative SH-SY5Y cell, expressing VirChR1 with (left) 10 mM HEPES pH 7.4, 110 mM NaCl, 2 mM MgCl$_2$ and (right) 110 mM L-arginine hydrochloride replacing NaCl in bath. Pipette solution during experiments was: 10 mM HEPES pH 7.4, 110 mM NaCl, 2 mM MgCl$_2$, 10 mM EGTA, illumination by LED (470 nm) lamp is indicated with light blue color. **c** Current–voltage dependences for $n = 1$ representative SH-SY5Y cell in 110 mM NaCl (red) and 110 mM L-arginine hydrochloride (blue). Currents are reproducible and typical to those in $n = 9$ experiments with other cells (and $n = 21$ experiments under slightly different NaCl concentrations varied from 110 mM to 140 mM). **d** Action spectrum of VirChR1 measured using equal photon fluxes (Sample size, $n = 18–20$). **e** Voltage-clamp records from $n = 1$ representative SH-SY5Y cell expressing VirChR1 in bath solution (left) 10 mM HEPES pH 7.4, 110 mM NaCl, 2 mM MgCl$_2$ and (right) in 80 mM CaCl$_2$ replacing NaCl in bath solution. **f** Current–voltage dependences for $n = 1$ representative SH-SY5Y cell in 110 mM NaCl (red) and 80 mM CaCl$_2$ (indigo) solutions. **g** Excitation recovery of photocurrent after a short pulse of nanosecond laser (500 nm) activation. Tau-off was measured in $n = 5$ independent experiments. Current–voltage dependences for $n = 1$ cell for different bath/pipette solution. Traces are shown for **h** bath solutions: 110 mM NaCl (red) and 110 mM KCl (green) (pipette solution is standard) and **i** pipette solution 110 mM L-arginine hydrochloride salt solution of pH 5.0 (bath solution is standard). Estimation of relative conductivities for different ions was done by fitting traces with Goldman-equation. **l** Current dependence on calcium concentration in bath solution measured at +80 mV (inflection point is at ~2.2 mM of calcium). For all electrophysiological recordings at $n = 1$ cell currents were reproducible in $n = 3–10$ independent experiments with other cells. No current averaging between cells was done, since different cells have significantly different protein expression levels. Data are presented as mean values ± SEM of current value under illumination in the cell measured.

the protein. To test the possible light-gated ion channeling by the VR1 group, we conducted electrophysiological studies of these proteins.

**Electrophysiology of VirChR1, a light-gated cation channel**. In order to identify the possible ion-conducting activity, we expressed human codon-optimized OLPVR1 and VirChR1 genes in SH-SY5Y human neuroblastoma cell line in the presence of all-*trans* retinal. Despite both proteins expressed well, they showed strong retention in the cytosol according to the fluorescence microscopy and electrophysiology data. To improve membrane trafficking and localization, we supplemented the proteins with C-terminal p2A self-cleavage peptide prior to fluorescent tag (see Methods for full details). This modification helped with VirChR1 localization and enabled us to analyze its photocurrents, however, OLPVR1 did not show significant improvements with this approach. Therefore, we characterized the ion-channeling activity of VirChR1 as a representative of the VR1 group 1.

Given the high sequence similarity of OLPVR1 and VirChR1 (Fig. 1b and Supplementary Fig. 2), we hereafter refer to the function of viral channelrhodopsins based on the data obtained for VirChR1. Whole-cell patch-clamp recordings revealed photocurrents generated by VirChR1 (Fig. 3b). Photocurrents were observed in a bath solution of 10 mM HEPES pH 7.4, 110 mM NaCl, 2 mM MgCl$_2$ and a pipette solution of 10 mM HEPES pH 7.4, 110 mM NaCl, 2 mM MgCl$_2$, 10 mM EGTA (hereafter both denoted standard). In response to continuous illumination by LED light ($\lambda_{max} = 470$ nm). Measuring photocurrents in different cells under standard conditions, we did not detect changes in kinetics or shifts of the reversal potential. For one representative neuroblastoma cell, photocurrent stabilized at 50 pA at 80 mV (Fig. 3b). The photocurrents varied in amplitude for different cells depending on the size of the cell and protein expression level, but the overall pattern remained the same. The photocurrent density in the neuroblastoma cells was $1.4 \pm 0.2$ pA/pF (mean ± std.dev, $n = 9$), which is comparably low compared to

the known channelrhodopsins, which might be due to a lower single channel conductance or due to less efficient expression of active molecules in the plasma membrane. The photocurrents showed a reversal potential, $U_{rev}$ of $25 \pm 6$ mV (mean $\pm$ std.dev, $n = 9$), indicating that light triggers a passive ion conductance by VirChR1. VirChR1 exhibits an action spectrum similar to those of typical rhodopsins, with the maximum sensitivity observed close to 500 nm (Fig. 3d). Tau-off for VirChR1 $\tau_{off} = 155 \pm 5$ ms (mean $\pm$ std.dev., $n = 5$) was directly determined using single-exponential fit of photocurrent recovery (Fig. 3g). Next, we performed ion substitution experiments to discriminate between possible cation and anion conductivity of the VirChR1 channel. First, we replaced the standard bath solution with 63 mM $Na_2HPO_4$/$NaH_2PO_4$ pH 7.4 and 2 mM $MgCl_2$, and found that this modification changed neither the magnitude nor the reversal potential of the photocurrent. By contrast, when replacing 110 mM NaCl in bath solution with 110 mM L-arginine hydrochloride, the inward current became immeasurably low resulting in a dramatic change in the current–voltage dependency (Fig. 3c). These results indicate that viral channelrhodopsins possess only cation-conducting activity.

**Unusual $Ca^{2+}$ sensitivity of VirChR1**. To evaluate the conductance of different cations by viral channelrhodopsins, we measured the voltage dependence of the photocurrent in different bath solutions. Replacing $Na^+$ with $K^+$ cations (Fig. 3h) in the bath solution yields an estimate of potassium permeability, $P_{K+} \approx 0.5 \cdot P_{Na+}$. To estimate $H^+$ permeability, we replaced the standard bath solution with 10 mM citric acid pH 5.0, 110 mM L-arginine hydrochloride, 2 mM $MgCl_2$ (Fig. 3i). Under these conditions, we observed full suppression of the photocurrent, which occurred, presumably, due to the protonation of the Schiff base proton acceptor (Fig. 2c). Therefore, instead of changing the standard bath solution, we replaced the pipette solution with 10 mM Citric Acid pH 5.0, 110 mM L-arginine hydrochloride, 2 mM $MgCl_2$, 10 mM EGTA, which preserved the photocurrent at a measurable level. Fitting the photocurrent data with the Goldman-Hodgkin-Katz equation estimated the $H^+$ permeability, $P_{H+} \approx 10^4$–$10^5 \cdot P_{Na+}$. Overall, VirChR1 shows ion selectivity comparable to those of CrChR2, namely, $P_{K+} \approx 0.5 \cdot P_{Na+}$ and $P_{H+} \approx 10^6 \cdot P_{Na+}$. Thus, group 1 of viral rhodopsins and chlorophyte cation channels are nearly similar with respect to the conductivity of monovalent ions.

Next, we tested whether VirChR1 was permeable for divalent cations, such as $Ca^{2+}$, similarly to CrChR2[13]. Strikingly, replacement of 110 mM NaCl in bath solution for 80 mM $CaCl_2$ completely abolished the photocurrent (Fig. 3e). The current–voltage dependences of VirChR1 in the presence and absence of $Ca^{2+}$ indicate that VirChR1 is completely impermeable for $Ca^{2+}$ cations (Fig. 3f), in contrast to the high $Ca^{2+}$ conductivity of CrChR2 (Supplementary Fig. 4). Importantly, VirChR1 is fully blocked for both inward and outward ionic fluxes when the concentration of $Ca^{2+}$ exceeds a certain threshold. To characterize the phenomena of VirChR1 inhibition by $Ca^{2+}$ ions, we measured the dependence of the average photocurrent at $+80$ mV voltage on the $CaCl_2$ concentration (Fig. 3j). We observed a sigmoid-like dependence with an inflection point at $K_{Ca2+} = 2.2$ mM $CaCl_2$, which is surprisingly close to the $Ca^{2+}$ concentration in the world ocean[37], and thus, suggestive of a functional role of viral rhodopsin inhibition by $Ca^{2+}$ ions. Taken together, our findings suggest that VirChR1 is a light-gated cation channel that conducts exclusively monovalent ions ($H^+$, $Na^+$, $K^+$) and is completely inhibited by divalent ions ($Ca^{2+}$) (Fig. 3a), with characteristic enzyme-substrate kinetics.

**VirChR1s are able to drive neural activity**. In order to verify the potential of viral channelrhodopsins for optogenetic stimulation, we transduced rat hippocampal neurons with adenoassociated virus (AAV-PHP.eB) carrying the VirChR1 gene under the control of the human synapsin I promoter. We used VirChR1 gene C-terminally fused to the Kir2.1 membrane trafficking signal, followed by a p2A self-cleavage peptide and Katushka fluorescent protein (see "Methods" for details). First, the experiments showed that VirChR1 protein was robustly expressed in hippocampal neurons (Fig. 4a). We found that VirChR1 with the N-terminal HA-FLAG tag, which we had used in the above experiments, caused major neuronal death, which made them impossible to measure with patch-clamp. In contrast, VirChR1 without the HA-FLAG tag expressed well and neurons were still viable enough for electrophysiological measurements. Second, patch-clamp experiments demonstrated firing in VirChR1-expressing neurons upon optical stimulation at 1 Hz (Fig. 4b). Due to the low photocurrent densities (in hippocampal neurons $0.5 \pm 0.2$ pA/pF measured at $-75$ mV $n = 11$, compared to $2.3 \pm 0.5$ pA/pF for CrChR2 and $8.0 \pm 0.4$ pA/pF for CatCh[38] (mean $\pm$ std.dev)) spikes were elicited with long latencies (Fig. 4c, d, latency of action potential peak is $119 \pm 6$ ms (mean $\pm$ std.dev) for this particular neuron). Optogenetic stimulation at higher rates failed to elicit spikes (Supplementary Fig. 5). However, neurons with higher photocurrents showed shorter spike latencies ($50 \pm 10$ ms, Supplementary Fig. 6). Furthermore, we observed action potentials arising after turning off the light, potentially due to the slow closing kinetics of the rhodopsin (Figs. 3g, 4c), which also led to double-spiking in other neurons (Supplementary Fig. 6). In conclusion, these results show that virus-mediated expression VirChR1s enables light-driven firing in neurons.

**Crystal structure of the viral rhodopsin OLPVR1**. To decipher the molecular mechanism of ion channeling, the structure of viral channelrhodopsin from group 1 is essential. One crystal structure of the $VirR_{DTS}$ (PDB ID: 6JO0[21]) was recently reported[21]. However, from the only available structure, it is not possible to distinguish the features of the entire group. Moreover, the ion-channeling activity of $VirR_{DTS}$ was not demonstrated in the original work[21]; thus structural insights of the channel of VirChR1s were not described. Here, we present a high-resolution structure of another VR1 protein, OLPVR1, at 1.4 Å resolution. It was crystallized with an *in meso* approach[39] similar to that used in our previous studies[3]. We obtained three different types of crystals. Type A rhombic crystals were grown at pH 8.0 using the monopalmitolein (MP) host lipid matrix and have the $P2_12_12$ space group with one protein molecule in the asymmetric unit. Type B hexagonal crystals were grown at pH 7.0 using a monoolein (MO) lipid matrix, have the P1 space group, and contain two protein molecules in the asymmetric unit. Type A and type B crystals resulted in the obtaining of the OLPVR1 structures at the resolution of 1.4 and 1.6 Å, respectively (Supplementary Table 1). OLPVR1 molecules are nearly identical in both structures (root mean square deviation (RMSD) less than 0.2 Å), so hereafter, we refer to the structure from type A crystals as it has the highest resolution. The crystal packing and examples of the electron density maps are shown in Supplementary Figs. 7 and 8.

The structure of the OLPVR1 protomer is composed of 7 transmembrane helices (TM1–7), connected by three intracellular and three extracellular loops (Figs. 1c, 5a). The OLPVR1 protein (residues 2–223), all-*trans* retinal (ATR) covalently bound to K204 (K257 in CrChR2[8]), 9 lipid molecules, and 107 water molecules are clearly resolved in the electron density maps (Supplementary Fig. 8). Despite the fact that only one OLPVR1

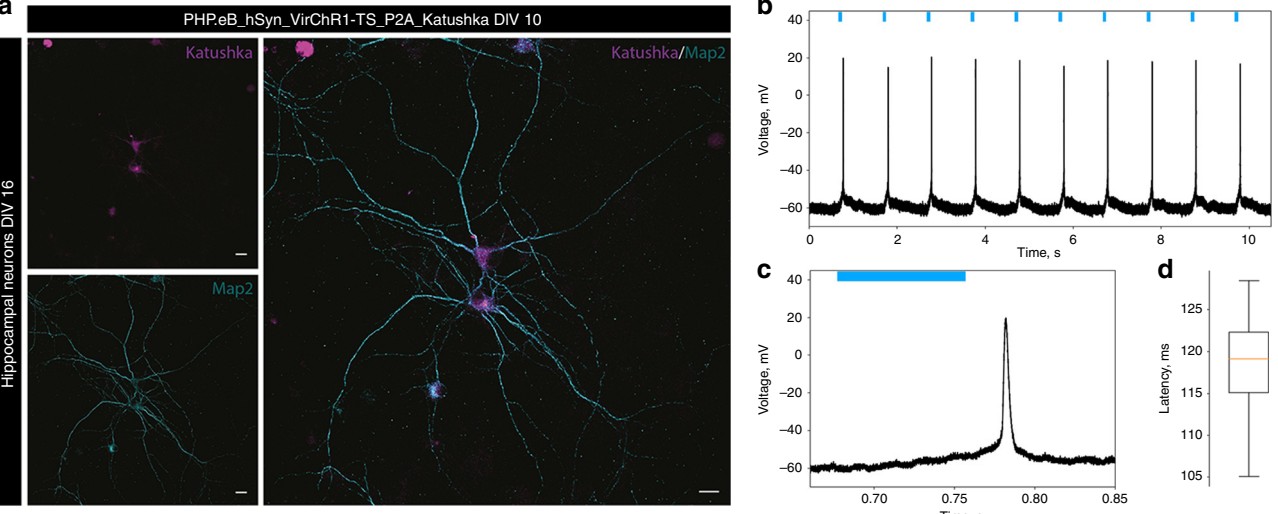

**Fig. 4 Neuronal application of VirChR1 for optogenetic activation. a** AAV transduction of primary hippocampal neurons at DIV10, intrinsic fluorescence of Katushka (magenta), and neuronal marker Map2 (cyan). Scale bars 10 μm. VirChr1 enables light-driven neuron spiking. Data presented in this figure refers to one representative neuron. The results showed in the microphotograph reproduced in $n = 6$ independent experiments. **b** Voltage trace showing depolarization and spikes of the neuron in response to the 1 Hz light pulse series, with 80 ms light pulses (green bars). **c** Expanded single spike induced by VirChR1 photoresponse. **d** Latencies distributions, when the neuron is illuminated with 80 ms light pulses. The box extends from lower quartile to upper quartile. The whiskers show the range of the latencies, green line is median latency, sample size, $n = 10$.

protomer is present in the asymmetric unit, the crystal packing of the protein shows that OLPVR1 could be organized into homodimers, similar to those of *Cr*ChR2[8,40]. These dimers might reflect the oligomeric state of the viral channelrhodopsin in the cell membrane (Supplementary Fig. 9). The interfacial interaction in the putative dimer occurs mainly in the TM4 helix and involves several non-conservative residues, namely E108-E108', Y111-Y111', F122-F122' with low surface interaction area (429 Å²) (Supplementary Fig. 9). An alternative configuration of OLPVR1 dimer predicted with GalaxyHomomer server[41] showed stronger interfacial interaction (1861 Å²). Therefore, the orientation of the OLPVR1 protomers in solution is unclear at the moment (Supplementary Fig. 9). Overall, the OLPVR1 backbone is tolerantly superimposed with that of the *Med12* proteorhodopsin (PDB ID: 4JQ6[42]) with RMSD of 2.1 Å, whereas the alignments with the *Hs*BR (PDB ID: 1C3W[26]) and *Cr*ChR2 (PDB ID: 6EID[8]) structures gives RMSD values of 4.3 and 3.7 Å, respectively (Fig. 1d, e and Supplementary Fig. 10). Interestingly, unlike other microbial rhodopsins, OLPVR1 architecture closely resembles the architecture of G-protein coupled receptors with TM3 helix protruding to the center of the protein. In particular, OLPVR1 aligns with bovine rhodopsin (PDB ID: 1U19[43]) with RMSD of 4.91 Å (Fig. 1f, g), with a high similarity among helices forming ion-conducting pathway (TM1–3 and TM7).

The OLPVR1 protomer has short extracellular loops, which sharply differentiates it from other channelrhodopsins that typically have large N- and C-terminal domains. Unlike in other microbial rhodopsins, helices TM3 and TM4 of OLPVR1 are connected by the loop containing the membrane-associated helix (ICL2 helix), which is composed of hydrophilic residues (Fig. 1c). Strikingly different from other rhodopsins, the intracellular parts of TM6 and TM7 helices of OLPVR1 are significantly moved apart from each other far enough to form a pore (Supplementary Fig. 11). The pore is located at about 8 Å from the cytoplasmic side of the lipid membrane border (Supplementary Fig. 11) and connects the inside of the protein with the groove on its surface, which leads further to the intracellular bulk. Surprisingly, in our structure, the groove and a part of the pore are occupied with a fragment of the lipid molecule, a host lipid of the crystallization

matrix (MP and MO in case of 1.4 and 1.6 Å structures, respectively), which is likely to be a crystallization artifact and is discussed in more details in the Supplementary Notes (Supplementary Fig. 11).

**Structure of the retinal binding pocket and Schiff base region.** The retinal cofactor is covalently attached to the conserved K204 residue in OLPVR1. $2F_o–F_c$ electron density maps at 1.4 Å reveal two alternative conformations of the retinal in the region of the β-ionone ring. At the same time, the configuration near the Schiff base in both of them is all-*trans* 15-*anti* (Supplementary Fig. 8). The retinal Schiff base (RSB) region of OLPVR1 is very similar to that of *Hs*BR (Supplementary Fig. 12c). Notably, the D76 and D200 side chains and water molecules w3, w10 and w11 in OLPVR1 (corresponding to D85, D212, w401, w402, and w406 in *Hs*BR, respectively) form an almost identical to *Hs*BR pentagon-like hydrogen bonds structure, which is important for proton transfer in light-driven proton pumps[44]. Moreover, the pentagon is similarly stabilized by T80 and R73 (T89 and R82 in *Hs*BR, respectively). In contrast, the pair of RSB counterions in *Cr*ChR2 is composed of E123 and D253, which, together with water molecules positioned in this region, presumably results in notably different stabilization of the RSB as compared to OLPVR1 (Supplementary Fig. 12c). The walls of the retinal pocket of OLPVR1 around the polyene chain are composed of several aromatic amino acids similar to those in *Hs*BR, namely W77, W178, and Y181 (Supplementary Fig. 12a). However, there are several important changes near the β-ionone ring, particularly L142, G182, and F185 instead of W138, P186, and W189 in *Hs*BR. These amino acids can potentially be candidates for mutation scanning to obtain red-shifted versions of VirChR1s, more suitable for practical applications, considering that OLPVR1 and *Hs*BR have retinal absorption maxima at 500 nm and 560 nm, respectively.

A characteristic feature of the retinal binding pocket of OLPVR1 (and, presumably, all viral channelrhodopsins) is the presence of highly-conserved and directly hydrogen-bonded residues T81 and N121 at the positions of the corresponding

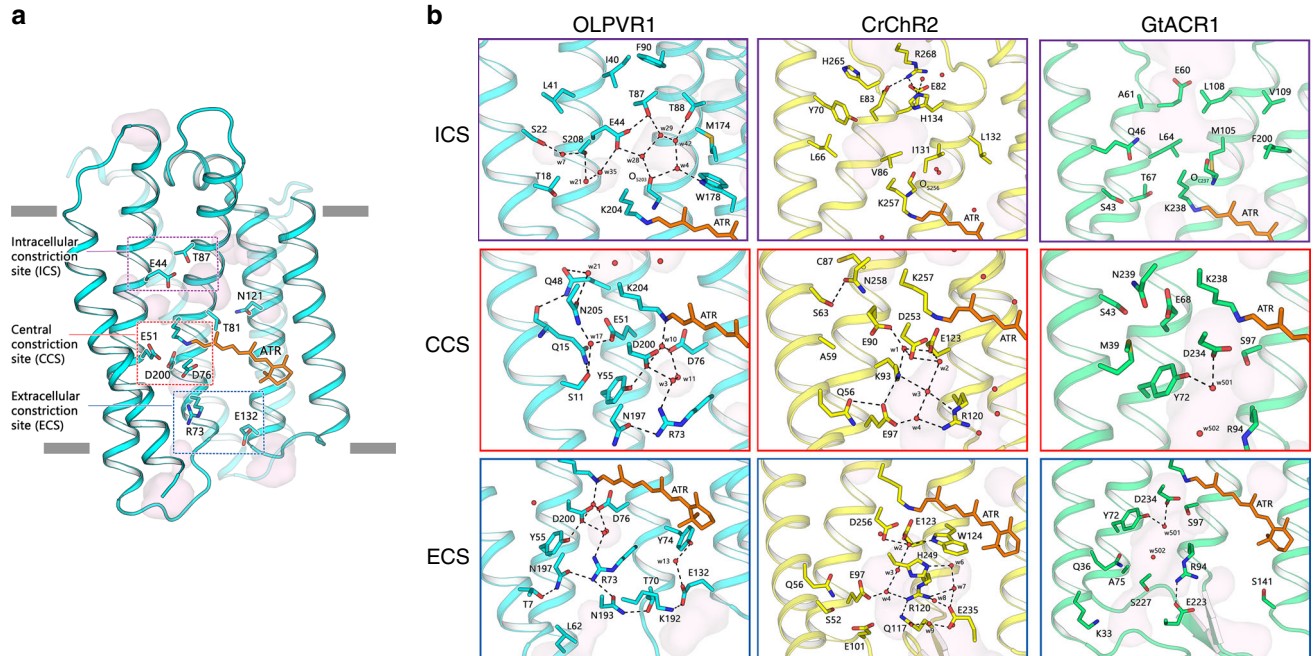

**Fig. 5 Organization of ion pathway constriction sites (CSs) in OLPVR1. a** Three CSs and cavities are forming the putative ion-conductive pathway in viral CCRs and highly conservative residues of OLPVR1. TM6 and TM7 helices are omitted for clarity. Membrane core boundaries were calculated using PPM server[103] and are shown with gray lines. **b** Magnified view of the CSs in OLPVR1 (left, present work), CrChR2 (middle, PDB ID: 6EID[8]) and (right, PDB ID: 6CSM[10]) structures, colored cyan, yellow and green, respectively. Water accessible cavities were calculated using HOLLOW[105] and are presented as a pink surface.

residues T80 and D115 in HsBR, and C128 and D156 in CrChR2 (Supplementary Fig. 12b). These pairs connect the middle parts of TM3 and TM4. Importantly, in the case of CrChR2, where C128 and D156 are interconnected by hydrogen bonds via a water molecule, the alteration of the pair dramatically affects the kinetics of the protein, and it was suggested that the pair is involved in the RSB reprotonation during photocycle[8,45,46].

**Organization of the OLPVR1 ion-conducting pathway.** Detailed analysis of the amino acid conservation in the VR1 group (Fig. 6a, b) shows that the majority of the conserved residues forms the interior of the protein differ from VR2 and PR groups (Fig. 6c) and are predicted to contribute to ion-channeling of VirChR1s. The structure suggests that the ion-conducting pathway of OLPVR1 is formed by TM1–3 and TM7 helices and is lined with several water-accessible cavities (Fig. 5a). A similar organization of VirR_DTS from the same VR1 group also supports this hypothesis (see Supplementary Note 1 and Supplementary Fig. 13). Unlike other channelrhodopsins, OLPVR1 lacks any prominent cavities in the extracellular part of the protein. Instead, it has a pore in the intracellular part, which ends up with a relatively large hydrophilic cavity inside the protein near the retinal (Fig. 5a, b). The cavity is filled with four water molecules (w4, w28, w29, w42) and surrounded by polar residues E44, T87, T88, and W178. Water molecules, together with the backbone oxygen of S203 residue, form a hydrogen bond pentagon (Fig. 5b) and may play a role in the hydration of cation during its translocation. A dense hydrogen bonding network involving water molecules and polar/charged residues protrude from the cavity almost to the extracellular bulk, only breaking in the central region near E51 residue. The putative ion pathway includes three constriction sites inside the protein (Fig. 5b). Each site (described in detail below) is comprise highly-conserved residues (Supplementary Fig. 2). The regions around the constriction sites are

almost identical in OLPVR1 and VirR_DTS[21] (see Supplementary Notes and Supplementary Fig. 13), and, therefore, we consider these to be a characteristic feature of VirChR1s and conjecture that they are also essential for the ion channeling of the entire VR1 group.

The intracellular constriction site (ICS) of the OLPVR1 is formed by the E44 side chain (Fig. 5b). It separates the large intracellular cavity from a polar region near T18, S22, Q48, N205, and S208 in the middle part of the protein, also containing three water molecules w7, w21 and w35. The E44 side chain is pointed towards the retinal, similarly to E122 in C1C2[40] (Supplementary Fig. 12), and is stabilized by hydrogen bonds with T87 and water molecules w20 and w35. Interestingly, unlike in CrChR2 and Chrimson[47], where the intracellular constriction sites (intracellular gates) are almost 14 Å far from the RSB and separated from it by the hydrophobic cavity, the ICS of OLPVR1 is located closer (9 Å) to the active center and is connected by extended hydrogen bonding network to the central constriction site (CCS) (Fig. 5b and Supplementary Fig. 14). Moreover, in CrChR2 and Chrimson, the ICSs are comprise tightly connected charged amino acids, completely blocking the flow of ions in the resting state (Fig. 5b and Supplementary Fig. 14). In these terms, the lack of compaction in the cytoplasmic region of OLPVR1 and the existence of the pore between TM6 and TM7 make the organization of the intracellular part of the protein closer to that of anion channel GtACR1[10,48], where the pore protrudes from the intracellular bulk almost down to the retinal without any constrictions (Fig. 5b). This might mean a different gating mechanism in the cytoplasmic part of OLPVR1 from other channelrhodopsins (Supplementary Fig. 15).

The CCS of the OLPVR1 includes the S11, Q15, E51, and N205 residues that are fully conserved among the members of the VR1 group and likely hinder the ion translocation pathway in the resting state (Fig. 6a, b). The core of the CCS is comprises S11–E51–N205 (S–E–N) triad, which is similar to S63-E90-

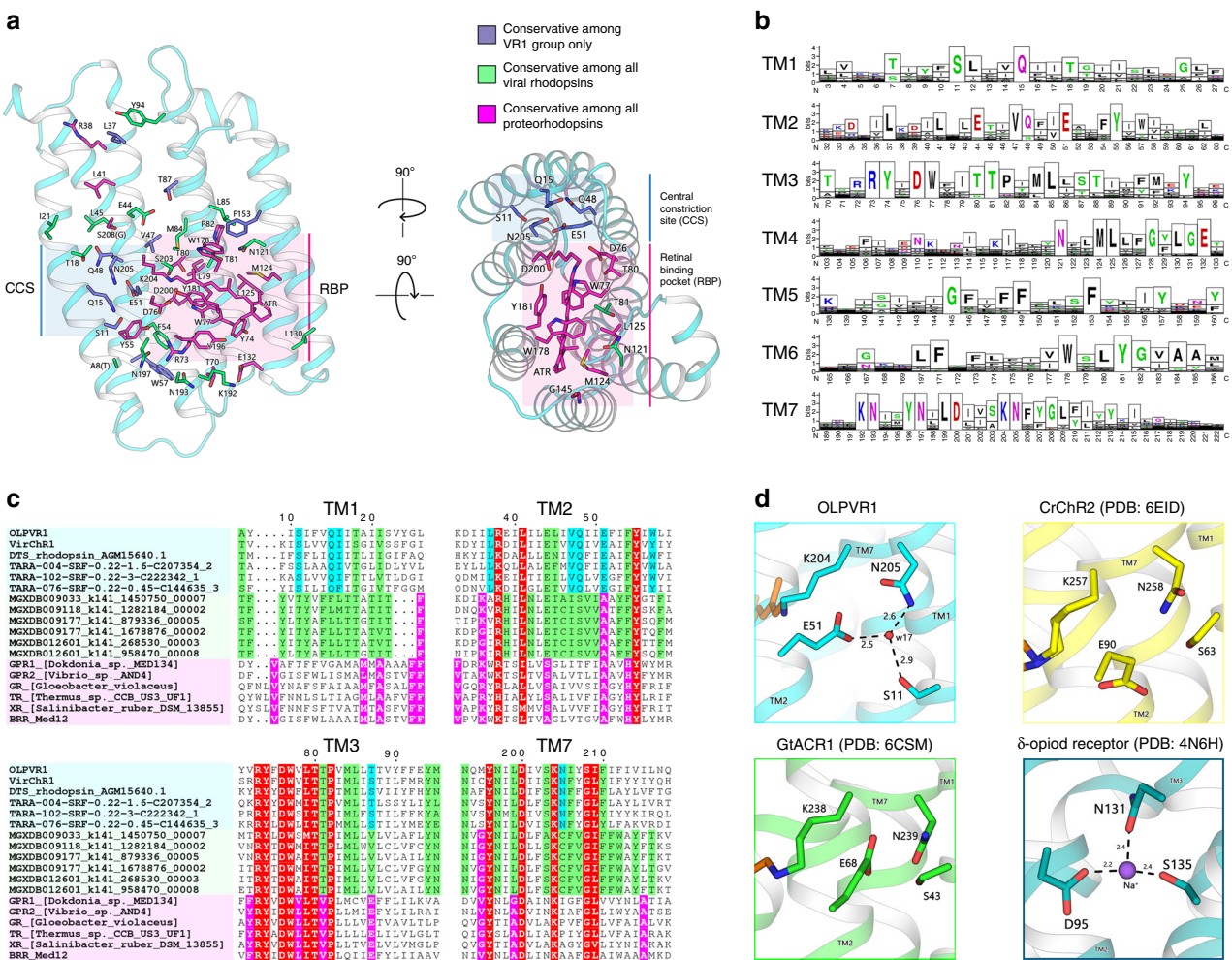

**Fig. 6 Conservativity analysis of viral rhodopsins. a** Structural overview of highly conservative (70% cutoff) residues among viral rhodopsins family, including both VR1 and VR2 groups ($n = 557$). OLPVR1 structure was used as a template of viral rhodopsin, viewed parallel from the membrane (left) and from the cytoplasm (right). Residues conservative among all proteorhodopsins (PRs) and residues exclusively conservative by viral rhodopsins are shown as sticks and colored magenta and dark blue, respectively. **b** Sequence logo of transmembrane helices (TM1–TM7) of viral rhodopsins family (both VR1 and VR2 groups) created using Weblogo sequence generator server[106]. **c** Sequence alignment of TM1, TM2, TM3 and TM7 helices of 6 representative sequences from the VR1 group, the VR2 group and marine PRs, colored light blue, green, and pink, respectively. Residues conservative among the VR1 group, the VR2 group, PRs, and both groups are colored cyan, green, magenta, and red, respectively. **d** The magnified view of the highly conservative S–E–N triad that comprises central constriction site (CCS) in OLPVR1 (top left), CrChR2 (PDB: 6EID[8], top right), and GtACR1 (PDB: 6CSM[10], down left). The sodium binding site formed by S–D–N triad in the human delta-opioid receptor (PDB: 4N6H[107], down right).

N258, S102-E129-N297 and S43–E68–N239 clusters of central gates (CGs) in CrChR2, C1C2, and GtACR1, respectively (Figs. 5b, 6d, and Supplementary Fig. 14). However, unlike in other channelrhodopsins, in OLPVR1, these residues are interconnected by a water molecule (w17), which thus define their orientation, even though no water accessible cavities were predicted in this region (Fig. 5b). Moreover, S11 is located one α-helix turn closer to the extracellular side than the corresponding serines in CrChR2, C1C2, and GtACR1 and is additionally stabilized by the hydrogen bond with the Q15 side chain (Figs. 5b, 6d). Q15 is a unique residue in OLPVR1 and has no analogs in other channelrhodopsins (Supplementary Fig. 3). In general, the organization of the S–E–N triad in OLPVR1 is closer to that of C1C2 and GtACR1, but not CrChR2 (Fig. 5b, Supplementary Fig. 14), where E90 side chain is pointed towards the RSB region and is a part of the extended hydrogen bond network in the extracellular part of the protein. Thus, unlike in CrChR2, in OLPVR1, the

CCS (and particularly E51) is not directly connected to the RSB region in the resting state (Fig. 5b).

The extracellular constriction site (ECS) of OLPVR1 includes the highly-conserved R73, E132, K192, N193, and N197 residues that are tightly interconnected by hydrogen bonds. In contrast to extracellular gates (ECGs) of CrChR2 and GtACR1, the R73 side chain is oriented towards the RSB, similarly to the analogous R82 in HsBR. However, it is stabilized by highly-conserved N193 and N197 residues in OLPVR1, while R82 in HsBR is stabilized mostly by the surrounding water molecules[26,44,49] (Fig. 5b, Supplementary Fig. 14). Also, unlike CrChR2, OLPVR1 lacks polar/charged amino acids in the region of E97 and E101 at the TM2 of CrChR2, which are substituted by leucines in the viral channelrhodopsin (Fig. 5b). Besides that, OLPVR1 possesses a more compact configuration of residues in the extracellular part than CrChR2, resulting in a more confined water-accessible cavities architecture of OLPVR1 (Supplementary Fig. 15). This might be another reason why virus rhodopsins from the VR1 group are not permeable for larger ions.

## Discussion

**Electrophysiology of VirChR1 and potential of VirChR1s for optogenetic applications.** Do all members of the VR1 function as ion channels? Taking into account all our data, high homology of VirChR1s, high conservation of functionally key amino acids, the high similarity of the structure of OLPVR1 to the recently described structure of VirR$_{DTS}$[21], another representative of the VR1, we suggest that this rhodopsin group (including VirR$_{DTS}$) should all comprise a distinct group of light-gated channels different from the known channelrhodopsins. It was recently reported that VirR$_{DTS}$ is a proton pump, based on the experiments with the *E. coli* plasma membrane expressing VirR$_{DTS}$, where the authors measured the pH changes in the *E. coli* suspension upon continuous light illumination[21]. We should note that such experiments are able to detect, in some cases, a proton-pumping activity; however, they are unable to identify a rhodopsin channel activity. Besides, only the observation of proton-pumping activity does not mean that the protein functions as an ion pump. Indeed, it has been shown that *Cr*ChR2 also pumps protons; it is the so-called "leaky proton pump"[50]. Moreover, this also is definitely the case of OLPVR1, studied in the present work (Fig. 2g). However, the experimental approach (ΔpH measurements) used in ref. [21] cannot prove the existence of ion channel properties. Therefore, by the direct electrophysiology approach, we detected and studied the ion channel properties, with HEK cells expressing VirR$_{DTS}$. The experiments showed that VirR$_{DTS}$ is also a light-gated channel (see Supplementary Notes and Supplementary Fig. 13). Indeed, predictably, the data showed photocurrents, which reverse their direction at approximately 0 mV, which is characteristic of rhodopsin ion channels but not of the ion pumps (Supplementary Fig. 14e).

Thus, taking into account all the data, high homology of VirChR1s, high conservation of functionally key amino acids, we suggest that the VirChR1s (including VirR$_{DTS}$ from giant PgV virus infecting the algae *Phaeocystis*) likely form a distinct group of compact light-gated Ca$^{2+}$-blocked channels, different from the known channelrhodopsins.

To the best of our knowledge, the exclusive conductivity for monovalent cations and its regulation by divalent cations have not been reported for any rhodopsin before. Recently, the cryptophyte alga *Guillardia Theta* has been reported to encompass genes of two different groups of channelrhodopsins, namely, anion-conducting channelrhodopsins with ~50% sequence identity to chlorophyte channelrhodopsins[9], and also a group with an architecture unconventional for channelrhodopsins, and containing the DTD motif that is characteristic of the archaeal proton pumps[18]. Consequently, these proteins were denoted BR-like cation channelrhodopsins (BCCRs). It has been shown that cation permeability of one of such proteins (*Gt*_CCR4) is reduced under high concentrations of Ca$^{2+}$ (more than 40 mM CaCl$_2$)[51]. However, such Ca$^{2+}$ concentrations are far beyond the physiological conditions. Moreover, even at 40 mM of Ca$^{2+}$, the channel permeability of Gt_CCR4 is not blocked and is about 20% of that at 2 mM Ca$^{2+}$[51]. It is unclear whether these results are biologically relevant. By contrast, VirChR1 is impermeable to divalent cations, and its channeling activity for monovalent cations is suppressed completely in the presence of a few mM of Ca$^{2+}$.

Despite the fact that *Cr*ChR2 seemingly exceeds VirChR1 performance in terms of optogenetics, the Ca$^{2+}$ impermeability is an important feature that separate VirChR1s from direct competition with chlorophyte channelrhodopsins. At the moment application of VirChR1 in optogenetics is limited by low photocurrent densities, poor plasma membrane localization and relatively slow kinetics. Nevertheless, we expect that VirChR1s may be useful for optogenetic applications because

VR1 family comprise more than 300 potential channelrhodopsins, some of which are likely to have improved plasma membrane localization and faster kinetics. Besides that, because VirChR1s would not interfere with important native Ca$^{2+}$-dependent processes in the cells, Ca$^{2+}$-impermeable channelrhodopsins could become valuable tools for Ca$^{2+}$-sensitive applications, for example, in cellular organelles like mitochondria, or in muscle cells and also to study processes in the brain, where optogenetic manipulation of synapses is advantageous and profitable[52–54]. In some cases, the activation of the slow light-gated channels can result in activation of voltage-gated calcium channels, that might be an issue for VirChR1 protein, however, faster VirChR1s would be able to overcome these limitations[55].

Another feature of VirChR1 is a non-zero negative photocurrent under symmetrical conditions at 0 mV (Fig. 3d, it also results in positive reversal potential). The same negative photocurrent has also been found when the channel activity was blocked by calcium (Fig. 3f). This suggests that inward-pumping activity could be responsible for this current. However, this explanation contradicts the results of pH measurements with OLPVR1 reconstituted to lipid vesicles. Additional work is required to resolve this discrepancy.

Also, the photocurrent of VirChR1 has an overshooting feature after turning the light off (Fig. 3b, e). Moreover, the photocurrent in the overshooting state tends to have reversal potential shifted towards zero. Although the causes of this effect remain unknown, we suggest that it might be explained by second-photon absorption during the measurements upon continuous illumination. The reversal potential shift in its turn can be explained by the change in the channeling-pumping ratio during the redistribution of proteins between photocycle states during overshooting.

**Distinct structural features of VirChR1s.** The comparison of the high-resolution structures of OLVPR1 and VirR$_{DTS}$ (see Supplementary Notes and Supplementary Fig. 13), together with the phylogenetic analysis of VR1, helped us to identify the distinctive structural features of the VirChR1s.

VirChR1s have proteorhodopsin-like architecture with short extracellular loops and share several structural features, such as membrane-associated ICL2 helix and an unconventional TM6-TM7 orientation. VirChR1s also share a set of highly-conserved residues that encompass the ion-conducting pathway and, by analogy with *Cr*ChR2, possess three consecutive constriction sites that are likely to be displaced in the open state of the channel (Supplementary Figs. 2, 15). Notably, VirChR1s lack the DC amino acid pair (C128 and D156 in *Cr*ChR2), which is replaced by an NT pair (T88 and N121 pair in OLPVR1) that is conserved in all VR1.

In contrast to VR2, VR1 members share the highly-conserved S–E–N triad (S11, E51, and N205) in the core of the protein, which likely plays the key role in the ion-conducting mechanisms of VirChR1s. At the same time, similar triads have been found in most of the known channelrhodopsins (Fig. 6d). In the case of OLPVR1 and VirChR1s in general, the S–E–N triad might also play a role in the inhibition of the protein activity by Ca$^{2+}$. We hypothesize that Ca$^{2+}$ binds near the triad and thus blocks the permeation of monovalent cations. This suggestion is based on the recently reported Na$^+$-bound O-state structure of the light-driven sodium pump KR2, described in[56]. Indeed, Na$^+$ has been shown to bind transiently in the core of the protein and is coordinated by the S70–N112–D116 triad, similar to the S11–E51–N205 triad of the CSS of OLPVR1. Given that Ca$^{2+}$ has similar coordination geometry to that of Na$^+$, it cannot be ruled out that Ca$^{2+}$ binds in the CSS of OLPVR1. Similarly to

KR2, where Na$^+$ binds to the protein only in the intermediate O-state, Ca$^{2+}$ binding also might occur not in the ground state of OLPVR1 but in the course of the protein photocycle. To clarify the details of Ca$^{2+}$ binding, as well as to elucidate the gating mechanism of VirChR1s, additional experiments should be performed; in particular, the structures of the intermediate states of OLPVR1 should be solved.

## Discussion

In this work, we demonstrate that VR1 rhodopsins are Na$^+$/K$^+$ selective light-gated ion channels that are inhibited by divalent cations. Viral channelrhodopsins are encoded together with retinal biosynthesis genes and widely distributed across the photic zone of World Ocean[21,57,58]. Large and especially giant viruses possess many auxiliary metabolic genes (AMG) that enhance the host metabolic functions and hence promote virus reproduction, without direct involvement in the virus replication processes[59–61]. By analogy with *Cr*ChR1 and *Cr*ChR2 channelrhodopsins from the chlorophyte alga *Chlamydomonas reinhardtii*[62], VirChR1s could be involved in the sensory activity of their hosts. They could also play the role of an additional ion channel to supplement and augment the host phototaxis systems. Photoexcitation of the channelrhodopsin receptors generates photoreceptor currents and membrane depolarization followed by activation of voltage-gated Ca$^{2+}$ channels triggering flagellar motion[62–65]. Due to their impermeability to Ca$^{2+}$ cations, viral channelrhodopsins could activate secondary Ca$^{2+}$-channels only by membrane depolarization, but not via biochemical amplification, as has been suggested for *Cr*ChR1 and *Cr*ChR2[62]. Thus, although the details of the function of viral channelrhodopsins remain to be explored, it appears likely that they enhance the light-induced motility of the host, and so boost anabolic processes required for virus reproduction.

However, some of the hosts infected by giant viruses apparently lack any light-sensory structures, and therefore, viral rhodopsins might be involved in other processes[21]. Many viruses encode small hydrophobic molecules, viroporins, that function as ion channels and size-limited pores and are able to permeabilize cellular membranes[66,67]. Viroporins can function as ion-conducting channels that open in either a voltage-dependent or in a voltage-independent manner, promoting virion assembly. Some of the viral channelrhodopsins might similarly augment virus budding using light energy[68,69]. Many phycodnaviruses, large viruses that belong to the NCLDV, encode channels, in particular, voltage-gated K$^+$-channels[70,71]. The channel proteins are essential for virus reproduction, but their specific functions have not been established. Thus, the role of rhodopsins in the reproduction of the viruses that encode them is part of the more general theme of the functions of diverse virus-encoded transport proteins that remain to be experimentally characterized[72,73].

## Methods

All experiments involving mouse materials were done in compliance with all relevant ethical regulations for animal testing and research. The study received ethical approval from the Lower Saxony State Office for Consumer Protection and Food Safety (LAVES). No experiments were randomized or blinded in this study. Sample sizes were determined based on prior literature and best practices in the field; no statistical methods were used to predetermine sample size.

**Metagenomic analysis**. Rhodopsins of the VR1 group were retrieved from metagenomic assembled contigs through combining similarity search, protein clustering, and Hidden Markov Models. Briefly, the first dataset of *bona fide* rhodopsins was retrieved from *TARA Ocean* metagenome-assembled contigs that were downloaded from ENA (https://www.ebi.ac.uk). In addition, a new assembly was performed for each sample starting by raw metagenomic reads. Sequencing reads were pre-processed using Trimmomatic[74] in order to remove low-quality bases (Phred quality score was set at 20, sliding windows of 4), and assembled using MEGAHIT[75] and the default parameters for generic metagenome assemblies.

Coding DNA sequences were predicted from contigs longer than 2 Kb using Prodigal[76], and annotated against the NR database of NCBI using Diamond[77]. *Bona fide* rhodopsins were selected by screening for different keywords related to rhodopsins that must be contained in the annotation, and filtered according to 7 transmembrane domains that were predicted using Phobius[78]. Selected proteins were next aligned using the R package Decipher[79], and alignments were used to infer a phylogenetic tree through FastTree 2[80] and using default parameters. Phylogenetic distances between nodes on the tree topology were considered for clustering *bona fide* rhodopsins into distinct clades, each of which was used to train a Hidden Markov Model (HMM). All HMMs were finally queried against *TARA Ocean* assembled contigs using HMMER version v3.1b2 (http://hmmer.org) and setting an e-value threshold of 1e-5. Proteins identified through HMMs were clustered at 100% identity using CD-HIT suite[81] to remove redundancy, and reduced to a total of 2584 Type-1 rhodopsins that were further analyzed.

The dataset of group I viral rhodopsins was constructed by searching the NCBI non-redundant protein sequence databases along with the TARA metagenomic sequences using BLSTP and TBLASTN. For the sake of clarity, for Supplementary Fig. 1, we used a reduced number of sequences. To obtain a representative set of 16 TARA metagenomic sequences with OLPVR1 and VirChR1 sequences included, we used the CD-HIT suite with default parameters and 60% identity cut-off.

**Sequence alignment and phylogenetic analysis**. Rhodopsin sequences were aligned using MUSCLE using UGENE software[82] with the default parameters. Type-1 rhodopsins were named according to their names in literature. The sequence alignment was created using ESPript3 online server[83]. Phylogenetic tree reconstruction was conducted by PHYLIP Neighbor Joining method using UGENE software[82] with the following parameters: Jones–Taylor–Thornton model, transition/transversion ratio = 2.0, no gamma distribution applied. Tree visualization was done using iTOL server[84]. GenBank accession numbers are additionally indicated.

**Cloning, expression, and purification**. The *E. coli* codon-optimized OLPVR1 and VirChR1 genes were synthesized commercially (Eurofins). The nucleotide sequence was optimized for *E. coli* expression using the GeneOptimizer software (Life Technologies). The gene, together with the 5′ ribosome-binding sites and the 3′ extensions coding additional LEHHHHHH* tag, was introduced into the pEKT expression vector (Novagen) via NdeI and XhoI restriction sites and verified by sequencing. VirChR1 protein was additionally supplemented with BRIL protein on the N-terminus of the protein, to improve protein folding and expression level[29]. Full plasmid and gene constructs, as well as primers used in this study can be found in Supplementary Tables 2–4. The proteins were expressed as described previously[7] with further modifications. *E. coli* cells of strain C41 (StabyCodon T7, Eurogentec, Belgium) were transformed with the expression plasmid. Transformed cells were grown in shaking baffled flasks in an autoinducing medium ZYP-5052 containing 50 mg/L kanamycin at 37 °C. When the OD$_{600}$ in the growing bacterial culture is 0.8–1.0 (glucose level < 10 mg/L), 10 μM all-*trans*-retinal (Sigma-Aldrich), and 1 mM isopropyl β-d-1-thiogalactopyranoside were added, the incubation temperature was reduced to 20 °C and incubated for 18 h. After incubation, cells were collected by centrifugation (5000 × g, 30 min) and disrupted in an M-110 P Lab Homogenizer (Microfluidics) at 20,000 p.s.i. in a buffer containing 20 mM Tris-HCl, pH 8.0 with 50 mg/L DNase I (Sigma-Aldrich). The membrane fraction of the cell lysate was isolated by ultracentrifugation at 90,000 × g for 1 h at 4 °C (Type 70 Ti Fixed-Angle Titanium Rotor, Beckmann). The pellet was resuspended in a buffer containing 20 mM NaH$_2$PO$_4$/Na$_2$HPO$_4$, pH 8.0, 0.1 M NaCl, and 1% n-dodecyl β-D-maltoside (DDM, Anatrace, Affymetrix) and stirred for 18 h for solubilization. The insoluble fraction was removed by ultracentrifugation at 90,000 × g for 1 h at 4 °C. The supernatant was loaded on a Ni-NTA column (Qiagen), and washed with a buffer containing 10 mM NaH$_2$PO$_4$/Na$_2$HPO$_4$, 150 mM NaCl, 30 mM imidazole, and 0.05% DDM buffer (pH 8.0). Elution of the protein was done in a buffer containing 10 mM NaH$_2$PO$_4$/Na$_2$HPO$_4$, 150 mM NaCl, 300 mM imidazole and 0.05% DDM (pH 8.0). The eluate was subjected to size-exclusion chromatography on a 20 ml Superdex 200i 10/300 GL column (GE Healthcare Life Sciences) in a buffer containing 10 mM NaH$_2$PO$_4$/Na$_2$HPO$_4$, pH 8.0, 150 mM NaCl, and 0.05% DDM. Protein-containing fractions with an A$_{280}$/A$_{500}$ absorbance ratio (peak ratio, p.r.) of lower than 1.5 were pooled and dialyzed against 100 volumes of 10 mM NaH$_2$PO$_4$/Na$_2$HPO$_4$, 150 mM NaCl, and 0.05% DDM (pH 8.0) buffer twice for 2 h to dispose of imidazole. The purified protein was concentrated for 40 mg/ml for crystallization.

**Reconstitution of the protein into lipid-based systems**. Phospholipids (azolectin from soybean, Sigma-Aldrich) were dissolved in CHCl$_3$ (Chloroform ultrapure, Applichem Panreac) and dried under a stream of N$_2$ in a glass vial. The solvent was removed by overnight incubation under vacuum. The dried lipids were resuspended in 100 mM NaCl buffer supplemented with 2% (w/v) sodium cholate. The mixture was clarified by sonication at 4 °C, and OLPVR1 was added at a protein/lipid ratio of 1:20 (w/w). The detergent was removed by 2 days stirring with detergent-absorbing beads (Amberlite XAD 2, Supelco). The mixture was dialyzed against 100 mM NaCl, (pH 7.0) buffer at 4 °C for 8 h to adjust the desired pH. The obtained liposomes were used for the measurement of pump activity with pH

electrode. The OLPVR1-containing nanodiscs were assembled using a standard protocol described elsewhere[85]. 1,2-dimyristoyl-sn-glycero-3-phosphocholine (DMPC, Avanti Polar Lipids, USA) and an MSP1D1 version of apolipoprotein-1 were used as a lipid and scaffold protein, respectively. The molar ratio during assembly was DMPC:MSP1D1:OLPVR1 = 100:2:3. The protein-containing nanodiscs were dialyzed against 100 volumes of 10 mM $NaH_2PO_4$/$Na_2HPO_4$, 100 mM NaCl (pH 7.5) buffer twice, and then subjected to size-exclusion chromatography on a 20 ml Superdex 200i 10/300 GL column (GE Healthcare Life Sciences) for detergent removal.

**Ion-trafficking assay with protein-containing liposomes**. The measurements were performed on 2 ml of stirred proteoliposomes suspension at 0 °C. OLPVR1- and LR/Mac- containing liposomes were prepared following the protocol described above. Liposomes were illuminated for 10 min with a halogen lamp (Intralux 5000-1, VOLPI) and then were kept in the dark for another 10 min. Changes in pH were monitored with a pH-meter (LAB 850, Schott Instruments). Some of the measurements were repeated in the presence of 30 μM of carbonyl cyanide m-chlorophenyl hydrazine (CCCP, Sigma-Aldrich) under the same conditions. We used purified LR protein incorporated in POPC:POPS liposomes (3:1) as a positive control. The codon-optimized *Leishmania tarentolae* LR gene was synthesized commercially (Eurofins). Full length LR protein was expressed in LEXSY expression system using protocols described for expression of *Cr*ChR2. Full protocol details are described in[8].

**pH titration**. To investigate the pH dependence of the absorption spectra of OLPVR1, about 6 μM protein was suspended in the titration buffer (10 mM citrate, 10 mM MES, 10 mM HEPES, 10 mM MOPS, 10 mM CHES and 10 mM CAPS). Then, the pH was changed by the addition of diluted or concentrated HCl or NaOH to obtain 0.5–0.7 pH change. The absorption spectra were measured with a UV-visible spectrometer (V-2600PC, Shimadzu).

**VirChR1 expression in mammalian cell lines**. The human codon-optimized OLPVR1, VirChR1 and DTS rhodopsin genes were synthesized commercially (Eurofins). The gene was cloned into the pcDNA3.1(−) vector bearing an additional membrane trafficking signal and YFP fluorescent protein (pCDNA3.1_-VirChR1_TS_YFP). The modified version of the proteins included a P2A self-cleaving peptide and Katushka fluorescent protein at the C-terminal part of the gene, and Hemagglutinin (HA) and Flag-tag peptides from the N-terminal part of the gene[86–88] (pCDNA3.1_HF_VirChR1_TS_P2A_Katushka). Full plasmid and gene constructs, as well as primers used in this study can be found in Supplementary Tables 2–4. The SH-SY5Y human neuroblastoma cells at a confluency of 80–90% were transfected with the plasmid and Lipofectamine LTX according to the manufacturer's protocol (Thermo Fisher Scientific). The cells were incubated under 5% $CO_2$ at 37 °C. After transfection (16–24 h), electrophysical experiments were performed.

**Electrophysiological recordings**. For the electrophysiological characterization of VirChR1, whole-cell patch-clamp recordings were performed (Scientifica LASU, Axon Digidata 1550A, Multiclamp 700B). Horizontal puller (Sutter Instrument CO, Model P-2000) was used for the fabrication of patch pipettes (borosilicate glass GB150F-8P, 3–6 MΩ). Experiments were conducted using SH-SY5Y cell line. Photocurrents were measured in response to LED light pulses with saturating intensity λ = 470 ± 20 nm (~3 mW/mm², and the corresponding photon flux was $7 \times 10^{15}$ photons/s/mm², assuming wavelength of emitted light to be 470 nm). For the action spectra, ultrashort nanosecond light pulses were generated by Brilliant Quantel using OPO Opotek MagicPrism for different wavelengths.

**Virus preparation**. For virus purification and transduction of the primary hippocampal neurons we used VirChR1 gene without N-terminal HA-FLAG tag (pAAV_hSyn_VirChR1_TS_P2_Katushka-WPRE_bGH), which was generated by site-directed mutagenesis using QuikChange II XL Site-Directed Mutagenesis Kit (Agilent) according to the manufacturer's instructions. Here, P2A is a self-cleaving peptide, TS is trafficking signal to the plasma membrane from inwardly rectifying potassium channel subunit Kir2.1[89], WPRE represents Woodchuck Hepatitis Virus (WHP) Posttranscriptional Regulatory Element, and bGH is polyadenylation element. The template for this mutagenesis was the original N-terminally HA-FLAG tagged pAAV construct similar to those used for SH-SY5Y experiments. AAVs were generated in HEK-293 T cells (ATCC) using polyethylenimine (25,000 MW, Polysciences, USA) transfection. Briefly, triple transfection of HEK-293T cells was performed using pHelper plasmid (TaKaRa/Clontech), plasmid providing viral capsid AAV-PHP.eB (PHP.eB was a gift from Viviana Gradinaru (Addgene plasmid # 103005; http://n2t.net/addgene:103005; RRID:Addgene_103005)), and plasmid providing the VirChR1 gene. The cell line was regularly tested for mycoplasma. We harvested viral particles 72 h after transfection from the medium and 120 h after transfection from cells and the medium. Viral particles from the medium were precipitated with 40% polyethylene glycol 8000 (Acros Organics, Germany) in 500 mM NaCl for 2 h at 4 °C and then after centrifugation at 4000 × *g* for 30 min combined with cell pellets for processing. The cell pellets were suspended in 500 mM NaCl, 40 mM Tris, 2.5 mM $MgCl_2$, pH 8, and 100 U/ml of salt-

activated nuclease (Arcticzymes, USA) at 37 °C for 30 min. Afterward, the cell lysates were clarified by centrifugation at 2000 × *g* for 10 min and then purified over iodixanol (Optiprep, Axis Shield, Norway) step gradients (15, 25, 40, and 60%) at 320,006 × *g* for 2.25 h. Viruses were concentrated using Amicon filters (EMD, UFC910024) and formulated in sterile phosphate buffered saline (PBS) supplemented with 0.001% Pluronic F-68 (Gibco, Germany). Virus titers were measured using AAV titration kit (TaKaRa/Clontech) according to the manufacturer's instructions by determining the number of DNase I-resistant vg using qPCR (StepOne, Applied Biosystems). Purity of produced viruses was routinely checked by silver staining (Pierce, Germany) after gel electrophoresis (NovexTM 4–12% Tris–glycine, Thermo Fisher Scientific) according to the manufacturer's instruction. The presence of viral capsid proteins was positively confirmed in all virus preparations. Viral stocks were kept at 80 °C until experimental day. The AAV production protocol is the same as previously described[90]. The titer for AAV-PHP.eB_hSyn_VirChR1-TS_P2A_Katushka_WPRE_bGH was $1.8 \times 10^{13}$ GC/ml.

**Hippocampal neuronal cultures, electrophysiology, and immunocytochemistry**. Primary cultured hippocampal neurons were prepared from embryonic E18/E19 Wistar rat pups. Isolated hippocampi were digested with Trypsin 0.25% and plated onto glass coverslips precoated with poly-D-lysine (25,000 cells per cm²). Hippocampal cultures were transduced at 10 DIV with AAV-PHP.eB virus carrying the VirChR1 gene (~$10^{10}$ GC/ml). Whole-cell patch-clamp measurements were performed at 16-17 DIV. The intracellular solution contained 129 mM K-gluconate, 10 mM HEPES, 10 mM KCl, 4 mM MgATP, 0.3 mM $Na_3$GTP, pH 7.2. Extracellular solution consisted of 140 mM NaCl, 1 mM $MgCl_2$, 2.5 mM KCl, 10 mM HEPES, 1.5 mM $CaCl_2$, 15 mM glucose, pH 7.4. 10 μM of NBQX and 50 μM of AP5 were added to the extracellular solution to block synaptic transmission. The illumination of cells was performed using 473 nm laser with saturation intensity (150 mW/cm²). All experiments were conducted at room temperature. Current-clamp recordings were performed with zero current injection. The electrophysiology data was processed using in-house Python 3 scripts (Jupyter Notebook). Hippocampal neurons used for immunocytochemistry were transduced at DIV 10 (final concentration $9 \times 10e9 − 1.8 \times 10e10$ gc) and after 5–6 days of incubation fixed by 4% paraformalaldehyde (PFA) and subjected further to immunostaining. Microtubule associated protein (Map2) was used as neuronal marker and immunolabeled by primary rabbit polyclonal Map2 antibody (Abcam, ab32454, 1:750). Secondary antibody against rabbit Map2 was Alexa Fluor goat anti-rabbit 647 IgG (Invitrogen, A21244, 1:750). VirChR1 was indirectly identified by intrinsic fluorescence of Katushka. All images were obtained with Leica scanning confocal microscope SP5 and processed using FIJI software. Final images were assembled in Adobe Illustrator software.

**Time-resolved absorption spectroscopy**. Excitation/detection systems were composed as such: Brilliant B laser with OPO Rainbow (Quantel Inc.) was used, providing pulses of 4-ns duration at 530-nm wavelength and an energy of 2 mJ per pulse. Samples (5 × 5–mm spectroscopic quartz cuvette; Hellma GmbH & Co.) were placed in a thermostated house between two collimated and mechanically coupled monochromators (LOT MSH150). The probing light (xenon arc lamp, 75 W, Hamamatsu) passed the first monochromator sample and arrived after a second monochromator at a photomultiplier tube (PMT) detector (R12829, Hamamatsu). The current-to-voltage converter of the PMT determines the time resolution of the measurement system of ca. 50 ns (measured as an apparent pulse width of the 5-ns laser pulse). Two digital oscilloscopes (Keysight DSO-x4022A) were used to record the traces of transient transmission changes in two overlapping time windows. The maximal digitizing rate was 10 ns per data point. Transient absorption changes were recorded from 10 ns after the laser pulses until full completion of the phototransformation. At each wavelength, 25 laser pulses were averaged to improve the signal-to-noise ratio. The quasilogarithmic data compression reduced the initial number of data points per trace (~32,000) to ~850 points evenly distributed in a log time scale giving ~100 points per time decade. The wavelengths were varied from 330 to 700 nm in steps of 10 nm using a computer-controlled step motor. Absorption spectra of the samples were measured before and after each experiment on a standard spectrophotometer (Avantes Avaspec 2048 L). Obtained datasets were independently analyzed using the multiexponential least-squares fitting by MEXFIT software[91]. The number of exponential components was incremented until the SD of weighted residuals did not further improve. After establishing the apparent rate constants and their assignment to the internal irreversible transitions of a single chain of relaxation processes, the amplitude spectra of exponents were transformed to the difference spectra of the corresponding intermediates in respect to the spectrum of the final state.

**Crystallization**. The crystals of OLPVR1 and O1O2 proteins were grown with an *in meso* approach[39], similar to that used in our previous works[3]. In particular, the solubilized protein (40 mg/ml) in the crystallization buffer was mixed with pre-melted at 50 °C monoolein (MO, Nu-Chek Prep) or monopalmitolein (MP, Nu-Chek Prep) in 3:2 ratio (lipid:protein) to form a lipidic mesophase. The mesophase was homogenized in coupled syringes (Hamilton) by transferring the mesophase from one syringe to another until a homogeneous and gel-like material was formed. 150 nl drops of a protein–mesophase mixture were spotted on a 96-well LCP glass

sandwich plate (Marienfeld) and overlaid with 400 nL of the precipitant solution by means of the NT8 crystallization robot (Formulatrix). The best crystals of OLPVR1 were obtained with a protein concentration of 20 mg/ml and 10 mM CaCl₂, 10 mM MgCl₂, 24% PEG 6000, 100 mM Tris (pH 8.2) for MP lipid and 10 mM CaCl₂, 10 mM MgCl₂, 24% PEG 550, 100 mM Tris (pH 8.2) for MO lipid (Hampton Research). The best crystals of O1O2 were obtained with a protein concentration of 20 mg/ml and 1.8 M Na₂HPO₄/KH₂PO₄ (pH 4.6). The crystals were grown at 22 °C and appeared in 1 to 4 weeks. Once crystals reached their final size, crystallization wells were opened, and drops containing the protein–mesophase mixture were covered with 100 μl of the respective precipitant solution. For data collection harvested crystals were incubated for 5 min in the respective precipitant solutions.

**Acquisition and treatment of diffraction data**. X-ray diffraction data of OLPVR1 were collected at the beamlines ID30b and ID23-1 of the ESRF, Grenoble, France, using a PILATUS 6M detector. The data collection at ESRF was performed using MxCube2 software. X-ray diffraction data of the O1O2 chimera were collected at the X06SA beamline of the SLS, Villigen, Switzerland, using EIGER 16M detector. Diffraction images were processed using XDS[92]. The reflection intensities were scaled using the AIMLESS software from the CCP4 program suite[93]. The reflection intensities of the highest-resolution data on OLPVR1 (1.4 Å) were also scaled using the Staraniso server[94] for the validation of the electron density maps quality. There is no possibility of twinning for the crystals. In all cases, diffraction data from one crystal was used. The data treatment statistics are presented in Supplementary Table 1.

**Structure determination and refinement**. Initial phases for the OLPVR1 structures were successfully obtained in the P2₁2₁2 and P1 space groups by molecular replacement (MR) using MOLREP[95] using the chain A of the 6SQG structure (OLPVRII protein) as a search model. Initial phases for O1O2 chimera were successfully obtained in the I121 space group by MR using the obtained structure of OLPVR1 as a search model. The initial MR models were iteratively refined using REFMAC5[96], PHENIX[97] and Coot[98]. The structure refinement statistics are presented in Supplementary Table 1.

**Molecular dynamics**. We used the refined 1.4 Å resolution crystallographic OLPVR1 structure for the initial conformation. All non-protein atoms except water were removed from the structure and the all-trans retinal molecule connected to the Lys were renamed using the retinol and retinal parameters for Charmm36 force field. The system then was prepared using Charmm GUI[99] input generator using the POPC lipid membrane and Tip3P water model. The resulting amount of lipids was 132, amount of water molecules −8901, amount of sodium ions −26, chlorine ions −23, overall system size was 48,350 atoms. Energy minimization and equilibration were performed in several steps with the gradual removal of spatial atomic constraints. The resulting simulation time was 1 μs (current time is 0.75 μs). Simulations were performed using velocity-rescale thermostat at 303.15 K and Parrinello-Rahman semi isotropic barostat with Gromacs 2018.4[100].

**Reporting summary**. Further information on research design is available in the Nature Research Reporting Summary linked to this article.

## Data availability

Data supporting the findings of this manuscript are available from the corresponding author upon reasonable request. A reporting summary for this Article is available as a Supplementary Information file. The protein coordinates and atomic structure factors have been deposited in the Protein Data Bank (PDB) under accession number PDB 7AKW (O1O2 mutant), PDB 7AKX (OLPVR1 in P1 space group), and PDB 7AKY (OLPVR1 in P21212 space group), respectively. Source data are provided with this paper.

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

## Acknowledgements

We thank C. Baeken for technical assistance. We thank D. Gerke for technical assistance during virus purification and S. Langer for technical assistance during hippocampal culture preparation. We acknowledge the Structural Biology Group of the European Synchrotron Radiation Facility (ESRF) for granting access to the synchrotron beamlines. We also acknowledge the Paul Scherrer Institut, Villigen, Switzerland for provision of synchrotron radiation beamtime at beamline X06SA of the SLS and would like to thank Dr. Anuschka Pauluhn for assistance. Electrophysiological experiments were supported by HGF-RSF grant (Helmholtz—RSF Joint Research Groups grant (RSF No. 19-44-06302)), the common program of Agence Nationale de la Recherche (ANR), France and Deutsche Forschungsgemeinschaft (DFG), Germany (ANR-15-CE11-0029-02/FA 301/11-1), by the DFG Research Unit FOR 2518 (*DynIon*, project P4 to JPM, MA 7525/1-1), and by funding from Frankfurt: Cluster of Excellence Frankfurt Macromolecular Complexes by the Max Planck Society (to E.B.) and by the Commissariat à l'Energie Atomique et aux Energies Alternatives (Institut de Biologie Structurale)–Helmholtz- Gemeinschaft Deutscher Forschungszentren (Forschungszentrum Jülich) Special Terms and Conditions 5.1 specific agreement. This work used the platforms of the Grenoble Instruct-ERIC center (ISBG; UMS 3518 CNRS-CEA-UJF-EMBL) within the Grenoble Partnership for Structural Biology (PSB). Platform access was supported by FRISBI (ANR-10-INBS-05-02) and GRAL, a project of the University Grenoble Alpes graduate school (Ecoles Universitaires de Recherche) CBH-EUR-GS (ANR-17-EURE-0003). Crystallographic experiments were funded by RFBR and CNRS according to the research project № 19-52-15017. This work was supported by the project ANR-19-CE11-0026. FRV was supported by grant VIREVO CGL2016-76273-P [AEI/FEDER, EU], (cofounded with FEDER funds). DZ was supported by the Russian Foundation for Basic Research project numbers 17-00-00164 and 20-34-90009. A.G., K.S. and M.K. were supported by Russian Foundation for Basic Research project numbers 17-00-00166 and 17-00-00167. V.G. was supported by the Ministry of Science and Higher Education of the Russian Federation (agreement # 075-00337-20-03, project FSMG -2020-0003). Work in Göttingen was funded by the European Research Council through the Advanced Grant 'OptoHear' to T. M. under the European Union's Horizon 2020 Research and Innovation program (grant agreement No. 670759) and the Deutsche Forschungsgemeinschaft (DFG, German Research Foundation) under Germany's Excellence Strategy - EXC 2067/1- 390729940.

## Author contributions

D.Z., A.A., and K.K. contributed equally and either has the right to list himself first in bibliographic documents. D.Z., T.B., D.B., and S.V. did molecular cloning, expressed and purified the proteins. D.V., D.S., and I.C. measured the photocycle kinetics and analyzed the corresponding data. D.Z. and R.A. crystallized the proteins. K.K. and D.Z. harvested crystals, processed the data and solved the structures with the help of A.P. A.A. and A.S._O. performed electrophysiology experiments under E.B., TMa., and M.V. supervision. E.S. performed electrophysiological experiments with DTS protein under A.O. supervision. T.Ma. directly supervised isolation of the hippocampal neurons. A.A. and E.P. performed site-directed mutagenesis and produced AAVs under direct supervision of V.R. V.R. and E.P. performed immunocytochemistry of hippocampal neurons and processed the images. D.Z., M.R., T.R., and Y.A. carried out the functional tests. G.A. and K.S. did the molecular dynamics experiments. R.R. and F.R.V. performed metagenomic search and identified new viral rhodopsin sequences. D.Z., N.Y., and E.K. analyzed the viral rhodopsins sequences and their possible biological role. V.G. initiated, designed and supervised the project and preparation of the manuscript. V.G., G.B., E.K, M.K., T.Mo., and E.B. planned and guided the work. D.Z., A.A., K.K., and V.G. analyzed the data and prepared the manuscript with the input from all other authors. D.W. and A.R. helped with preparation of the revised version of the manuscript.

## Competing interests

The authors declare no competing interests.

## Additional information

**Peer review information** *Nature Communications* thanks Satoshi Tsunoda and other, anonymous, reviewers for their contributions to the peer review o this work. Peer review reports are available.

Dmitrii Zabelskii [1,2,3,26], Alexey Alekseev[1,2,3,4,26], Kirill Kovalev[1,2,3,4,5,26], Vladan Rankovic [6,7], Taras Balandin[1,2], Dmytro Soloviov [3,8,9], Dmitry Bratanov[1,2], Ekaterina Savelyeva[10,11,12], Elizaveta Podolyak[3], Dmytro Volkov [1,2], Svetlana Vaganova[1,2], Roman Astashkin[3,5], Igor Chizhov[13], Natalia Yutin[14],

Maksim Rulev[1,2,15], Alexander Popov[15], Ana-Sofia Eria-Oliveira [5], Tatiana Rokitskaya [16], Thomas Mager[6,17], Yuri Antonenko[16], Riccardo Rosselli [18,19], Grigoriy Armeev[20], Konstantin Shaitan [20,21], Michel Vivaudou[5,22], Georg Büldt[3], Andrey Rogachev[3,8], Francisco Rodriguez-Valera [3,18], Mikhail Kirpichnikov[20,23], Tobias Moser [6,7,17], Andreas Offenhäusser [10], Dieter Willbold [1,2,24], Eugene Koonin [14], Ernst Bamberg[3,25] & Valentin Gordeliy [1,2,3,4,5✉]

[1]Institute of Biological Information Processing (IBI-7: Structural Biochemistry), Forschungszentrum Jülich GmbH, Jülich, Germany. [2]JuStruct: Jülich Center for Structural Biology, Forschungszentrum Jülich GmbH, Jülich, Germany. [3]Research Center for Molecular Mechanisms of Aging and Age-related Diseases, Moscow Institute of Physics and Technology, Dolgoprudny, Russia. [4]Institute of Crystallography, University of Aachen (RWTH), Aachen, Germany. [5]Institut de Biologie Structurale (IBS), Université Grenoble Alpes, CEA, CNRS, Grenoble, France. [6]Institute for Auditory Neuroscience and InnerEarLab, University Medical Center Göttingen, Göttingen, Germany. [7]Auditory Neuroscience and Optogenetics Laboratory, German Primate Center, Göttingen, Germany. [8]Joint Institute for Nuclear Research, Dubna, Russia. [9]Institute for Safety Problems of Nuclear Power Plants, NAS of Ukraine, Kyiv 03680, Ukraine. [10]Institute of Biological Information Processing (IBI-3: Bioelectronics), Forschungszentrum Jülich GmbH, Jülich, Germany. [11]Laboratory of Functional Materials and Devices for Nanoelectronics, Moscow Institute of Physics and Technology, Dolgoprudny, Russia. [12]Center of Shared Research Facilities, Moscow Institute of Physics and Technology, Dolgoprudny, Russia. [13]Institute for Biophysical Chemistry, Hannover Medical School, Hannover, Germany. [14]National Center for Biotechnology Information, National Library of Medicine, National Institutes of Health, Bethesda, MD, USA. [15]European Synchrotron Radiation Facility, Grenoble, France. [16]Belozersky Institute of Physico-Chemical Biology, Lomonosov Moscow State University, Moscow, Russia. [17]Cluster of Excellence "Multiscale Bioimaging: from Molecular Machines to Networks of Excitable Cells" (MBExC), University of Göttingen, Göttingen, Germany. [18]Evolutionary Genomics Group, Departamento de Producción Vegetal y Microbiología, Universidad Miguel Hernández, San Juan de Alicante, Spain. [19]Department of Marine Microbiology and Biogeochemistry, Royal Netherland Institute for Sea Research (NIOZ), and Utrecht University, Den Burg, The Netherlands. [20]Biological Faculty, M. V. Lomonosov Moscow State University, Moscow 119991, Russia. [21]N. N. Semenov Institute of Chemical Physics, Russian Academy of Sciences, Moscow 119991, Russia. [22]Laboratories of Excellence, Ion Channel Science and Therapeutics, 06560 Valbonne, France. [23]M. M. Shemyakin-Yu. A. Ovchinnikov Institute of Bioorganic Chemistry, Russian Academy of Sciences, Moscow 117997, Russia. [24]Institut für Physikalische Biologie, Heinrich Heine University Düsseldorf, Düsseldorf, Germany. [25]Max Planck Institute of Biophysics, Frankfurt am Main, Germany. [26]These authors contributed equally: Dmitrii Zabelskii, Alexey Alekseev, Kirill Kovalev. ✉email: valentin.gordeliy@ibs.fr

