## [Peer Review File · Nature Communications]

REVIEWER COMMENTS

Reviewer #1 (Remarks to the Author):

This manuscript has been revised to address the previous reviewers' concerns. The experiments added during the revision do make this study more complete, but there are several points that are still not very well done and would benefit from a more rigorous approach. In particular, the authors are still adhering by their tendency to present anecdotal data (n = 1 neuron in Fig. 4, for example) and making statements that are not supported with robust statistics. Once these are corrected, I believe that this manuscript will be suitable for publication in nature communications.

Specific points:

- In the electrophysiology data, the authors state an n = 9 ("The photocurrents showed a reversal potential, U_{rev} of 25 ± 6 mV (mean \pm std.dev, n = 9)") and yet the figure legend (Fig. 3e) claims to show a typical response that was observed in 30 additional cells. Why are these 30 cells not included in the population data being reported? Why are the current-voltage plots in Fig. 3f, i-I showing only individual cells and not the population average?
- The additional photocurrent associated with switching off the light pulse seems to merit more investigation. The authors claim that this might be a "second-photon effect". However, it seems like this current has a different reversal potential than the initial photocurrent induced by light ON – it seems to reverse around -50mV. The simple conclusion would be that this current has not only a unique photocycle but also a different ion selectivity. Could the authors plot the I-V curves for this current separately from the light onset-associated current? This can be a supplementary figure, but it seems important enough to report.
- The authors make a major point about the VirChRs being superior to CrChR2 due to their reduced Ca⁺⁺ conduction. However, is Ca⁺⁺ conduction through ChR2 really a major issue? Based on previous works (e.g. Zhang and Oertner, Nat Methods 2007) the majority of calcium influx after ChR2 activation results from recruitment of voltage-gated ion channels and not the endogenous Ca⁺⁺ permeability of ChR2. However, slow ChRs tend to induce greater calcium influx than faster ones since they trigger more prolonged opening of VGCCs. In that sense, the VirChRs seem to belong to the "slow ChR" category (although the off kinetics are not measured or reported in the manuscript) and could therefore trigger elevated VGCC activation compared with faster ChRs.
- "We found that VirChR1 without the N-terminal HA-FLAG tag, which we had used in the above experiments, expressed well in neurons and did not affect their viability" – this statements requires some quantitative evidence. Although VirChR1 expressed better than OLVPR1, it still appears to be substantially present in intracellular compartments (Fig. 4a) and therefore might cause cell health issues when overexpressed. The authors should compare expressing vs. non-expressing cells (or at least ChR2-expressing ones) and record resting membrane potential, membrane capacitance, membrane resistance etc. to support the above claim.
- On that topic, Fig. 4 which presents the evidence for the potential utility of VirChR1 for neuronal applications, contains data taken from only one neuron. This is highly uncommon in the field and requires some more rigorous investigation. Although a few additional traces are shown in Extended Data Fig. 5, quantifying spike fidelity in neurons as a function of light power, pulse length and frequency would greatly improve this aspect of the paper. Otherwise, I would suggest that the authors remove this anecdotal data from the manuscript and focus on the biophysics and structural results.
- Extended data fig. 3: "Key amino acids involved in ion channeling are colored blue and purple" – I see only red color in this alignment.

- Extended data fig. 4: why is the plot for Chr2 only showing the "CaCl₂" condition and not the NaCl one as well? The omission is peculiar. Also, the contents of the extracellular and intracellular solutions in these experiments are not provided.

Reviewer #2 (Remarks to the Author):

The manuscript by Zebelskii et al. describes functional and structural studies of two rhodopsins from marine viruses, OLPVR1 and VirChR1 for the first time.

The biggest achievement of the study is the determination of the crystal structure of OLPVR1 in which they found a striking difference in TM6-7 arrangement compared to other microbial rhodopsins. The authors reveal it forms a functional dimer. Also, they identify a putative channel pore (ion pathway) which includes three constriction sites in the protein.

The authors also elucidated interesting characteristics of the pH-dependent protonation/deprotonation of the retinal Schiff-base by spectroscopic experiments. Also, by electrophysiological method, the authors observed light-induced ion-transport which is diminished by Ca²⁺.

In addition, they reveal that VirRDTS, which was previously shown to work as a proton pump by other researches, functions as a light-gated ion channel (Extended data Fig 13). The results in the manuscript indicate that VirChR1s form a distinct group of cation channels.

Furthermore, the author demonstrated that VirChR1 can induce neuronal spiking by illumination when tested in hippocampal neurons. This suggests the potential of VirChR1 as an optogenetics tool, although the time resolution might not be high as Chr2.

In the previous manuscript submitted to Nature, much of the presented data were preliminary and premature. However, I find significant improvement in the quality of data in the current version. I appreciate the author's effort.

This manuscript would be of high interest to a broad community of scientists and thus justify publication in Nature communications after some corrections.

The followings are comments.

1. In Fig. 2i, pKa of the spectral shift was determined as 3.50 for VirChR1 and 4.8 for OLPVR1. However, the results are not mentioned and discussed in the text. Explain the reason for the difference between VirChR1 and OLPVR1.

2. Page 7 lines 14-16 "In response to continuous illumination by LED light ($\lambda_{max} = 470$ nm), the photocurrents of VirChR1 revealed partial desensitization and decayed to a stationary level."

Due to the poor quality of the traces (Fig. 3d), "partial desensitization" is invisible. To me, some traces seems to slowly increase to a stationary level during illumination without desensitization. The sentence should be modified.

3. In Fig. 3b, photocurrent response upon a flash laser was shown. But the results are not described and discussed in the text. Also, kinetics (tau-off) could be determined from the trace.

4. In Fig. 6b and d, amino acid sequence alignment are shown. However, these are not described in the text.

5. Fig.1 legends are mixed up except for Fig. 1a. Please correct.

6. Fig. 2 panel order.

In the text, Fig. 2f appears first, followed by Fig.2e, h, a, b, c, g.. This reduces the readability very

much. Please rearrange the panel order.

7. Fig. 3 panel order. Same as Fig. 2.

In the text, the panels appear Fig. 3d, g, c, e, f, j, e, h, and then i. It would be very difficult for readers to follow. Please rearrange the panel order.

In addition, Fig. 3d and 3g are essentially the same measurements (in different cells?). Fig. 3g could be deleted to avoid confusion.

Reviewer #3 (Remarks to the Author):

The manuscript is a well revised and extended version of a previous manuscript submitted to Nature. The authors satisfactorily addressed my criticism related to the manuscript submitted to Nature and I have no further critique.

The manuscript addresses a broad readership and is suitable for publication in Nature Communications.

Reviewer #1 (Remarks to the Author):

This manuscript has been revised to address the previous reviewers' concerns. The experiments added during the revision do make this study more complete, but there are several points that are still not very well done and would benefit from a more rigorous approach. In particular, the authors are still adhering by their tendency to present anecdotal data ($n = 1$ neuron in Fig. 4, for example) and making statements that are not supported with robust statistics. Once these are corrected, I believe that this manuscript will be suitable for publication in nature communications.

First of all, we would like to thank the reviewer for valuable suggestions and helpful criticism. We addressed the comments of the reviewer and made the corresponding changes in the manuscript. We also performed additional experiments to address properly a comment.

Specific points:

In the electrophysiology data, the authors state an $n = 9$ (“The photocurrents showed a reversal potential, U_{rev} of 25 ± 6 mV (mean \pm std.dev, $n = 9$)”) and yet the figure legend (Fig. 3e) claims to show a typical response that was observed in 30 additional cells. Why are these 30 cells not included in the population data being reported? Why are the current-voltage plots in Fig. 3f, i-l showing only individual cells and not the population average?

We performed experiments with VirChR1 under different conditions. Some of them were similar. In this particular case, we observed reproducible current traces in 30 cells under high sodium concentrations. However, we used different concentrations of NaCl (from 110 mM to 140 mM). Therefore, we averaged traces from only part ($n = 9$) of the cells, which were measured under exactly the same conditions. To avoid misunderstanding we modified the caption of Figure 3 by adding the following sentence (page 43, line 6): “Currents are reproducible and typical of those in 9 experiments with other cells (and 21 more cells under slightly different NaCl concentrations varied from 110 mM to 140 mM), illumination by LED (470 nm) lamp is indicated with light blue color.”.

The additional photocurrent associated with switching off the light pulse seems to merit more investigation. The authors claim that this might be a “second-photon effect”. However, it seems like this current has a different reversal potential than the initial photocurrent induced by light ON – it seems to reverse around -50mV. The simple conclusion would be that this current has not only a unique photocycle but also a different ion selectivity. Could the authors plot the I-V curves for this current separately from

the light onset-associated current? This can be a supplementary figure, but it seems important enough to report.

We appreciate the suggestion of the referee. Although we supposed that the second photon effect might explain the overshooting effect, we agree that the reversal potential being calculated in the overshoot state is different from that in the steady-state. However, we conducted the corresponding experiments under symmetrical ionic conditions. Therefore, only the presence of multiple open states with different ion selectivity cannot explain the reversal potential shift.

In fact, a pure channel cannot have non-zero reversal potential under symmetrical ionic conditions in both steady and off states. Therefore, the reversal potential shift could be only explained by the change in the pumping/channeling ratio. The change of the ratio in turn might be explained by the presence of multiple open states with different net ion permeability. However, there might be other explanations.

For example, the dependence of certain photocycle time constants (such as τ_{off}) on voltage is common among channelrhodopsins¹. Such dependencies result in changes in voltage-current in the closing-state compared to the steady-state. Thus, it could also lead to a reversal potential shift if a protein is not a pure channel.

Taking into account the comment of the reviewer we modified the corresponding text of the discussion as follows (page 16, line 29):

“Another feature of VirChR1 is a non-zero negative photocurrent under symmetrical conditions at 0 mV (Figure 3d, it also results in positive reversal potential). The same negative photocurrent has also been found when the channel activity was blocked by calcium (Figure 3f). This suggests that inward-pumping activity could be responsible for this current. However, this explanation contradicts the results of pH measurements with OLPVR1 reconstituted to lipid vesicles. Additional work is required to resolve this discrepancy.

Also, the photocurrent of VirChR1 has an overshooting feature after turning the light off (Figure 3b, e). Moreover, the photocurrent in the overshooting state tends to have reversal potential shifted towards zero. Although the causes of this effect remain unknown, we suggest that it might be explained by second-photon absorption during the measurements upon continuous illumination. The reversal potential shift in its turn can be explained by the change in the channeling-pumping ratio during the redistribution of proteins between photocycle states during overshooting.”

- The authors make a major point about the VirChRs being superior to CrChR2 due to their reduced Ca⁺⁺ conduction. However, is Ca⁺⁺ conduction through ChR2 really a major issue? Based on previous works (e.g. Zhang and Oertner, Nat Methods 2007) the majority of calcium influx after ChR2 activation results from recruitment of voltage-gated ion channels and not the endogenous Ca⁺⁺ permeability of ChR2. However, slow ChRs tend to induce greater calcium influx than faster ones since they trigger more prolonged opening of VGCCs. In that sense, the VirChRs seem to belong to the “slow ChR” category (although the off kinetics are not measured or reported in the manuscript) and could therefore trigger elevated VGCC activation compared with faster ChRs.

Thanks for the comment. First of all, we do not consider that VirChR1 is superior to CrChR2, however, we agree that we did not emphasize it clearly enough in the previous version, so we reworked on the corresponding section of the text.

In fact, we consider that VirChR1 might be used complementarily to CCRs in applications, where its Ca²⁺ impermeability might be an advantage over other channelrhodopsins, in such processes as optogenetic manipulation of synapses and optogenetic control of muscle cells and cell organelles as mitochondria.

To avoid this misunderstanding, we modified the following parts of the manuscript (page 16, line 15) :
”Despite the fact that CrChR2 seemingly exceeds VirChR1 performance in terms of optogenetics, the Ca²⁺ impermeability is an important feature that separates VirChR1s from direct competition with chlorophyte channelrhodopsins. At the moment application of VirChR1 in optogenetics is limited by low photocurrent densities, poor plasma membrane localization and relatively slow kinetics. Nevertheless, we expect that VirChR1 may be useful for optogenetic applications because the VR1 family comprises more than 300 potential channelrhodopsins, some of which might have improved plasma membrane localization and faster kinetics. Besides that, because VirChR1s would not interfere with important native Ca²⁺-dependent processes in the cells, Ca²⁺-impermeable channelrhodopsins could become valuable tools for Ca²⁺-sensitive applications, for example, in cellular organelles like mitochondria, or in muscle cells and also for the study of processes in the brain, where optogenetic manipulation of synapses is advantageous and profitable²⁻⁴. ”

Second, we agree that there can be delayed calcium influx into neurons due to the activation of VGCCs. However, this limitation is only relevant to “slow ChRs”, and therefore VirChR1s with faster kinetics would not have this problem. In fact, there are already multiple examples of successful modifications of CCRs that significantly improved channel kinetics^{5,6}. Therefore, we believe that activation of elevated

VGCC might be considered a problem only for the VirChR1 protein, but not for all the VirChR1 family. Besides that, we measured the tau off time for VirChR1 protein ($\tau_{\text{off}} = 155 \pm 5$ ms; mean \pm std.dev., n = 5) and added it to Figure 3g and to the main text (page 8, line 7): “Tau-off for VirChR1 $\tau_{\text{off}} = 155 \pm 5$ ms (mean \pm std.dev., n = 5) was directly determined using single-exponential fit of photocurrent recovery (Figure 3g).

Taking into account the comment of the reviewer, we added the following sentence to the text of the manuscript (page 16 , line 25): “In some cases, the activation of the slow light-gated channels may result in activation of voltage-gated calcium channels, that might be an issue for VirChR1 protein, however, faster VirChR1s would be able to overcome these limitations⁷.”

“We found that VirChR1 without the N-terminal HA-FLAG tag, which we had used in the above experiments, expressed well in neurons and did not affect their viability” – this statements requires some quantitative evidence. Although VirChR1 expressed better than OLVPRI, it still appears to be substantially present in intracellular compartments (Fig. 4a) and therefore might cause cell health issues when overexpressed. The authors should compare expressing vs. non-expressing cells (or at least ChR2-expressing ones) and record resting membrane potential, membrane capacitance, membrane resistance etc. to support the above claim.

Regarding this question, we were able to obtain sufficient photocurrents only with VirChR1 supplemented with additional self-cleavage peptide at C-term (p2A peptide⁸) that allowed separate expression of channelrhodopsin and fluorescent tag. In order to make this point clear for the readers we additionally included the following sentences to the corresponding parts of the text (page 9, line 19): “We used VirChR1 gene C-terminally fused to the Kir2.1 membrane trafficking signal, followed by a p2A self-cleavage peptide and Katushka fluorescent protein (see Methods for details).”; (page 7, line 12): “Despite the fact that both proteins expressed well, they showed strong retention in the cytosol according to the fluorescence microscopy and electrophysiology data. To improve membrane trafficking and localization we supplemented the proteins with C-terminal p2A self-cleavage peptide prior to fluorescent tag (see Methods for full details). This modification helped with VirChR1 localization and enabled us to analyze its photocurrents, however, OLVPRI did not show significant improvements with this approach.” Besides that, we updated corresponding part in Methods section (page 23, line 9).

Due to the cleavage of fluorescent tag, the microscopy data (Figure 4a) do not show the exact localization of the VirChR1, but rather provide a brief estimation of protein expression level. We stated that VirChR1 without the N-terminal HA-FLAG was successfully expressed in hippocampal neurons, so we were able to conduct experiments with those neurons. In contrast, expression of VirChR1 with the N-terminal HA-FLAG led to the death of most neurons, so we could not collect any meaningful data. Therefore, we addressed in the text the discrepancy in the influence on neurons viability between those two constructs. Nevertheless, we agree that this section can be improved, so, to avoid misunderstandings, we modified the corresponding part of the text as follows (page 9 line 22): “We found that VirChR1 with the N-terminal HA-FLAG tag, which we used in the above experiments, caused major neuronal death, which made it impossible to measure them with patch-clamp. In contrast, VirChR1 without the HA-FLAG tag expressed well and neurons were still viable enough for electrophysiological measurements.”.

On that topic, Fig. 4 which presents the evidence for the potential utility of VirChR1 for neuronal applications, contains data taken from only one neuron. This is highly uncommon in the field and requires some more rigorous investigation. Although a few additional traces are shown in Extended Data Fig. 5, quantifying spike fidelity in neurons as a function of light power, pulse length and frequency would greatly improve this aspect of the paper. Otherwise, I would suggest that the authors remove this anecdotal data from the manuscript and focus on the biophysics and structural results.

We performed experiments of neuronal firing with several cells and showed that VirChR1 is, in principle, capable of driving action potentials. But due to the low photocurrent densities in the current experiments, we found that latencies are highly dependent on the expression level of functional VirChR1. Therefore, it makes no sense to average latencies of neurons with different levels of VirChR1 expression. The representative examples of VirChR1 neuron spikes are additionally shown in Extended Data Figure 5. Besides that, we demonstrate that in the case of higher photocurrents, VirChR1 could elicit spikes with higher frequencies and lower latencies, which is stated in the text “However, neurons with higher photocurrents showed shorter spike latencies (50 ± 10 ms, Extended Data Figure 6).”.

Extended data fig. 3: “Key amino acids involved in ion channeling are colored blue and purple” – I see only red color in this alignment.

The description of Extended Data Figure 4 was corrected, the word ‘purple’ was removed as it does not correspond to the current version of this figure (page 52, line 6).

Extended data fig. 4: why is the plot for ChR2 only showing the “CaCl₂” condition and not the NaCl one as well? The omission is peculiar. Also, the contents of the extracellular and intracellular solutions in these experiments are not provided.

Taking into account the reviewer comment we conducted the corresponding additional experiment with ChR2 protein in SH-SY5Y cells. The results are now presented in Extended Data Figure 4b, the corresponding extracellular and intracellular solutions are mentioned to the Figure caption (page 53, line 1).

Reviewer #2 (Remarks to the Author):

The manuscript by Zabelskii et al. describes functional and structural studies of two rhodopsins from marine viruses, OLPVR1 and VirChR1 for the first time. The biggest achievement of the study is the determination of the crystal structure of OLPVR1 in which they found a striking difference in TM6-7 arrangement compared to other microbial rhodopsins. The authors reveal it forms a functional dimer. Also, they identify a putative channel pore (ion pathway) which includes three constriction sites in the protein. The authors also elucidated interesting characteristics of the pH-dependent protonation/deprotonation of the retinal Schiff-base by spectroscopic experiments. Also, by electrophysiological method, the authors observed light-induced ion-transport which is diminished by Ca²⁺. In addition, they reveal that VirRDTS, which was previously shown to work as a proton pump by other researches, functions as a light-gated ion channel (Extended data Fig 13). The results in the manuscript indicate that VirChR1s form a distinct group of cation channels. Furthermore, the author demonstrated that VirChR1 can induce neuronal spiking by illumination when tested in hippocampal neurons. This suggests the potential of VirChR1 as an optogenetics tool, although the time resolution might not be high as ChR2. In the previous manuscript submitted to Nature, much of the presented data were preliminary and premature. However, I find significant improvement in the quality of data in the current version. I appreciate the author’s effort. This manuscript would be of high interest to a broad community of scientists and thus justify publication in Nature communications after some corrections.

We thank the reviewer for appreciation of our work and constructive suggestions. We made the necessary corrections to address the questions put by the reviewer in the updated version of the manuscript. Please find below our detailed answers to reviewer comments.

The followings are comments.

1. In Fig. 2i, pKa of the spectral shift was determined as 3.50 for VirChR1 and 4.8 for OLPVR1. However, the results are not mentioned and discussed in the text. Explain the reason for the difference between VirChR1 and OLPVR1.

We agree with the reviewer comment, so we extended the corresponding section in the main text of the updated version with the following text, aimed to clarify this question (page 5, line 18):

“To characterize photochemical properties of viral channelrhodopsins, we expressed C-terminally his-tagged OLPVR1 and VirChR1 proteins in *E.coli* and purified them using a combination of Ni-NTA and size exclusion chromatography methods (see Methods for details). The VirChR1 protein was additionally supplemented with the BRIL protein on the N-terminus of the protein, to improve the expression level of the protein⁹. Both OLPVR1 and VirChR1 show absorption spectra sensitive to blue light with λ_{\max} of 500 nm and 507 nm respectively at pH 7.5 (Figure 2a). Similar to the VirR_{DTS} rhodopsin¹⁰, HsBR¹¹ and proteorhodopsins¹², OLPVR1 and VirChR1 undergo a ~ 30 nm spectral red-shift under acidic conditions, associated with the protonation of retinal chromophore counterion (Figure 2b,c). The Schiff base region of VirChR1s is reminiscent of those in light-driven proton pumps, such as HsBR, suggesting that D76 in OLPVR1 (D80 in VirChR1) acts as counterion, as in HsBR (Figure 2d). In order to estimate the pKa values we plotted the absorption maximum values against buffer pH and fitted the data by the Henderson-Hasselbalch equation with a single pKa (Figure 2e). The resulting pKa values for OLPVR1 (pKa = 4.8) and VirChR1 (pKa = 3.5) are in good agreement with pKa = 3.6, previously reported for VirR_{DTS} rhodopsin¹⁰. The one-unit difference between OLPVR1 and VirChR1 pKa values might be possibly explained by the difference in relative positions of the TM1-3 and TM7 helices and the difference in the neighboring to counterion residues, such as I50 and L79 in OLPVR1, which are replaced with V53 and I83 in VirChR1 (Extended Data Figure 2).” Besides that, we updated corresponding part in Methods section (page 21, line 6).

2. Page 7 lines 14-16 “In response to continuous illumination by LED light ($\lambda_{\max} = 470$ nm), the photocurrents of VirChR1 revealed partial desensitization and decayed to a stationary level.”

Due to the poor quality of the traces (Fig. 3d), “partial desensitization” is invisible. To me, some traces seems to slowly increase to a stationary level during illumination without desensitization. The sentence should be modified.

We agree with the comment, so we removed this sentence about protein desensitization, to avoid a misinterpretation.

3. In Fig. 3b, photocurrent response upon a flash laser was shown. But the results are not described and discussed in the text. Also, kinetics (tau-off) could be determined from the trace.

We performed additional analysis and determined the tau-off. The result is shown in the Figure 3g and in the sentence added to the main text (page 8, line 7): “Tau-off for VirChR1 $\tau_{\text{off}} = 155 \pm 5$ ms (mean \pm std.dev., n = 5) was directly determined using the single-exponential fit of photocurrent recovery (Figure 3g).”

4. In Fig. 6b and d, amino acid sequence alignment are shown. However, these are not described in the text.

We modified the main text sections with the corresponding references to Figure 6b and 6c, that were missing in the previous version of the manuscript (page 13 line 1): “Detailed analysis of the amino acid conservation in the VR1 group (Figure 6a,b) shows that the majority of the conserved residues form the interior of the protein different from the VR2 and PR groups (Figure 6c) and are predicted to contribute to ion-channeling of VirChR1s.” Figure 6d was referenced, so the text was not further modified.

5. Fig.1 legends are mixed up except for Fig. 1a. Please correct.

Figure 1 description was rearranged accordingly (page 40, line 5).

6. Fig. 2 panel order.

In the text, Fig. 2f appears first, followed by Fig.2e, h, a, b, c, g.. This reduces the readability very much. Please rearrange the panel order.

We rearranged Figure 2 subplots according to their appearance in the corresponding section of the main text (page 42, line 1). Besides that, the section was extended in order to improve readability (the corresponding text is quoted in answer to comment 1).

7. Fig. 3 panel order. Same as Fig. 2. In the text, the panels appear Fig. 3d, g, c, e, f, j, e, h, and then i. It

would be very difficult for readers to follow. Please rearrange the panel order. In addition, Fig. 3d and 3g are essentially the same measurements (in different cells?). Fig. 3g could be deleted to avoid confusion.

Figure 3 was changed in order to clarify the difference between subfigures, Figure 3 caption and its references in the main text were updated correspondingly (page 43, line 1). The data in figures 3d and 3g were collected on different cells to show the exact effect of bath solution replacement by perfusion. As we stated in the manuscript, the photocurrents varied in amplitude for different cells depending on the size of the cell and protein expression level, therefore we decided to present examples of the data collected on the representative cell, but the effect was reproducible over multiple experiments with slightly different amplitudes.

Reviewer #3 (Remarks to the Author):

The manuscript is a well revised and extended version of a previous manuscript submitted to Nature. The authors satisfactorily addressed my criticism related to the manuscript submitted to Nature and I have no further critique.

The manuscript addresses a broad readership and is suitable for publication in Nature Communications.

We thank the reviewer for the appreciation of our work.

References:

1. Mager, T. *et al.* High frequency neural spiking and auditory signaling by ultrafast red-shifted optogenetics. *Nat. Commun.* **9**, 1750 (2018).
2. Lin, J. Y. A user's guide to channelrhodopsin variants: Features, limitations and future developments. *Experimental Physiology* (2011) doi:10.1113/expphysiol.2009.051961.
3. Lin, J. Y. *et al.* Optogenetic inhibition of synaptic release with chromophore-assisted light inactivation (CALI). *Neuron* (2013) doi:10.1016/j.neuron.2013.05.022.
4. Tritsch, N. X., Granger, A. J. & Sabatini, B. L. Mechanisms and functions of GABA co-release. *Nat. Rev. Neurosci.* (2016) doi:10.1038/nrn.2015.21.

5. Mager, T. *et al.* Improved Microbial Rhodopsins for Ultrafast Red-Shifted Optogenetics. *Biophys. J.* **114**, 669a (2018).
6. Bedbrook, C. N. *et al.* Machine learning-guided channelrhodopsin engineering enables minimally invasive optogenetics. *Nat. Methods* **16**, 1176–1184 (2019).
7. Zhang, Y.-P. & Oertner, T. G. Optical induction of synaptic plasticity using a light-sensitive channel. *Nat. Methods* **4**, 139–141 (2007).
8. Wang, Y., Wang, F., Wang, R., Zhao, P. & Xia, Q. 2A self-cleaving peptide-based multi-gene expression system in the silkworm *Bombyx mori*. *Sci. Rep.* **5**, 16273 (2015).
9. Chun, E. *et al.* Fusion Partner Toolchest for the Stabilization and Crystallization of G Protein-Coupled Receptors. *Structure* **20**, 967–976 (2012).
10. Needham, D. M. *et al.* A distinct lineage of giant viruses brings a rhodopsin photosystem to unicellular marine predators. *Proc. Natl. Acad. Sci.* (2019) doi:10.1073/pnas.1907517116.
11. Balashov, S. P. Protonation reactions and their coupling in bacteriorhodopsin. *Biochim. Biophys. Acta BBA - Bioenerg.* **1460**, 75–94 (2000).
12. Huber, R. *et al.* pH-Dependent Photoisomerization of Retinal in Proteorhodopsin[†]. *Biochemistry* **44**, 1800–1806 (2005).

REVIEWERS' COMMENTS

Reviewer #1 (Remarks to the Author):

The authors have adequately addressed my criticisms and I support the publication of this manuscript in Nature Communications. I wish to congratulate the authors on their important work, and thank them for the hard work of preparing and revising this manuscript.